# Dietary flavonoids may improve insulin resistance: NHANES, network pharmacological analyses and in vitro experiments

Xiaohui Sui[1☯], Yanhong Liu[2☯], Junde Zhao[1], Zuocheng Wang[3], Guiju Zhang[2*]

**1** Shandong University of Traditional Chinese Medicine, Jinan, Shandong, China, **2** Affiliated Hospital of Shandong University of Traditional Chinese Medicine, Jinan, Shandong, China, **3** Australian National University Research School of Biology, Canberra, Australia

☯ Xiaohui Sui and Yanhong Liu are co-first authors.
* 13685316296@163.com

## Abstract

### Background

Insulin resistance (IR) constitutes the pivotal pathological link underlying numerous metabolic diseases and represents a paramount global health challenge. Flavonoids have demonstrated beneficial effects on diseases such as cancer and hypertension through their anti-inflammatory and antioxidant activities. However, the relationship between dietary flavonoid intake and IR prevalence remains unclear.

### Methods

The cross-sectional study utilized population data from the 2007–2010 and 2017–2018 National Health and Nutrition Examination Surveys (NHANES). IR assessment employed the metabolic score for insulin resistance (METS-IR). The relationship between dietary flavonoid intake and IR underwent analysis by weighted generalized linear regression and weighted restricted cubic splines (RCS). The mechanism of flavonoid action on IR was studied by network pharmacology and molecular docking, and verified in vitro. Since the free form compound did not exist, we conducted subsequent experiments using the stable salt form Cyanidin Chloride.

### Results

Our findings revealed statistically significant negative associations between Anthocyanidins (P<0.0001) and Flavanones (P<0.001) and IR. Meanwhile, relative to the minimal concentration group, the low concentration group (β=−1.39, P=0.02) and the moderate concentration group (β=−2.84, P=0.001) of total flavonol intake had the potential to reduce METS-IR. RCS showed that total flavonoids, Anthocyanidins, Isoflavone, Flavan-3-ols, Flavanones, Flavones and Flavonols had a nonlinear correlation with IR. In vitro experimentation corroborated these findings: after adding

**Data availability statement:** All data generated or analysed during this study are included in this published article (and its Supplementary Information files).

**Funding:** This work was funded by the Science and Technology Co-construction Project of the Science and Technology Department of the National Administration of Traditional Chinese Medicine (GZY-KJS-SD-2024-082, awarded to Guiju Zhang).

**Competing interests:** The authors declare that they have no known competing financial interests or personal relationships that could have appeared to influence the work reported in this paper.

anthocyanidins and hesperidin supplementation, glucose uptake was significantly restored in the IR group, ameliorating IR.

## Conclusion

Our study highlights the importance of appropriately increasing dietary flavonoid intake when improving IR.

---

### 1. Introduction

Insulin resistance (IR) denotes a pathological state wherein insulin target tissues – encompassing the liver, fat and muscle – exhibit markedly diminished responsiveness to insulin signals [1]. Its core manifestations encompass impaired glucose uptake and utilization, insufficient inhibition of liver gluconeogenesis, and uncontrolled lipolysis of adipose tissue [2]. It assumes pivotal significance in the development of various metabolic diseases such as type 2 diabetes mellitus (T2DM) and metabolic syndrome (MetS) [3–4]. Epidemiological studies demonstrate that the incidence of IR is escalating globally, with approximately 51% of the population manifesting varying degrees of IR [5]. Meanwhile, existing evidence indicates that IR constitutes a principal catalyst for cardiovascular diseases (CVD), non-alcoholic fatty liver disease (NAFLD), polycystic ovary syndrome (PCOS), and certain neurodegenerative disorders [6]. IR can be precipitated by multiple factors, among which obesity represents the paramount and modifiable risk factor. Additionally, genetic factors, autoimmune diseases, lipid metabolism disorders and environmental factors can all induce IR [7]. Oxidative stress, chronic inflammation, mitochondrial dysfunction and abnormal signal transduction constitute its common pathological mechanism [8]. Current treatments for IR include lifestyle intervention, behavioral therapy and drug therapy; however, these treatments retain certain limitations [9–11]. Therefore, there exists a pressing necessity to identify effective interventions to treat IR and related metabolic diseases.

Flavonoids are natural polyphenols found in fruits and vegetables [12]. Contemporary investigations have progressively elucidated their remarkable therapeutic potential in cancer, hypertension and CVD [13–15]. In addition, various subclasses of flavonoids exhibit key biological activities related to the pathogenesis of metabolic diseases, encompassing obesity, T2DM, and hyperlipidemia. For instance, a diet rich in anthocyanins can regulate fasting blood glucose and glycated hemoglobin levels, improving the prognosis of T2DM patients [16–17]. Luteolin attenuates blood glucose and insulin levels in T2DM rats and improves IR by inhibiting the activity of α -glucosidase [18]. Comprehensive catechins assimilation exhibits inverse correlations with hyperlipidemia manifestation frequencies [19]. Isorlicorice improves IR by up-regulating the expression of insulin signal-related genes in the liver and muscles, while increasing the expression of thermogenic genes, enhancing energy expenditure, and improving body fat mass [20]. Concurrently, flavonoids have potent anti-inflammatory and antioxidant activities, functioning not exclusively as antioxidants to clear reactive oxygen species (ROS), but also constraining

regulatory enzymes involved in the inflammatory process [21]. Such explorations intimate that flavonoid compounds may serve as promising candidates for dietary interventions against IR, whereby a flavonoid-rich diet can help prevent certain metabolic diseases.

The investigation endeavored to elucidate associations between dietary flavonoid intake and IR from a clinical perspective using publicly available data from the National Health and Nutrition Examination Survey (NHANES). Network pharmacology analysis with molecular docking explored mechanistic pathways, culminating in experimental validation to identify novel therapeutic targets for the prevention and treatment of IR.

## 2. Materials and methods

### 2.1. Data sources and study population

Our study used publicly available data from the NHANES. NHANES is a nationwide, complex and multi-stage probabilistic sample survey conducted by the National Center for Health Statistics (NCHS) to provide health and nutrition data on the U.S. population [22]. This research utilized comprehensive data and detailed information regarding NHANES data collection available via the website (http://www.cdc.gov/nchs/nhanes.html). This study has been reviewed and approved by the NCHS Ethics Review Board, and all participants signed informed consent. The report was prepared in accordance with the Strengthening the Reporting of Observational Studies in Epidemiology (STROBE) guidelines for cross-sectional studies (Supplementary material STROBE-nut_checklist).

To explore the potential link between flavonoid intake and IR, this study acquired data from the NHANES database spanning three cycles in 2007–2008, 2009–2010, and 2017–2018, including a total of 29,940 participants. Following rigorous exclusion criteria, including exclusion of individuals under 20 years of age and pregnant individuals, participants with missing metabolic score for insulin resistance (METS-IR) data, participants with missing flavonoid intake data, and participants with missing other covariates [such as education level, family poverty income ratio (PIR), smoking, etc.], 3,564 participants were ultimately included in the study (Fig 1).

### 2.2. Assessment of flavonoid intakes

All dietary intake data systematically collected in NHANES were encoded using the United States Department of Agriculture (USDA) Food and Nutrient Database for Dietary Studies (FNDDS) database. The nutrient intake of each food was then calculated using the USDA Automated Multiple-Pass Method and correlated with specific flavonoid values from the USDA Survey Food Code Flavonoid Value Database (Flavonoid Database) [23–24]. We collected data pertaining to dietary flavonoid intake from the Flavonoid Database for 2007−2010 and 2017−2018. The database contains detailed information on 29 flavones, delineated into six categories: isoflavones, anthocyanins, flavonols, flavan-3-ols, flavanones, and flavones [25]. This study leveraged the average of flavonoid intake from 2 24-hour dietary recalls to define the final flavonoid intake. It merits emphasis that this intake was ascertained exclusively through dietary sources and excludes intake of flavonoid supplements or medications.

### 2.3. Assessment of METS-IR

In this study, the METS-IR served as the primary outcome variable for evaluating individual IR status. The calculation formula is as follows: METS-IR = Ln [(2 × fasting glucose)+fasting triglycerides] × body mass index/ [Ln (high-density lipoprotein cholesterol) [26]. Among them, fasting glucose, fasting triglycerides and HDL data are obtained from the 'Laboratory Data' module, which is measured by an automatic biochemical analyzer. Body mass index (BMI) is calculated by dividing weight (kg) by the square of height (m). Height and weight measurements are extracted from the 'Body Measures' section, where weight quantification occurs via Toledo electronic scale and height quantification occurs via fixed rangefinder.

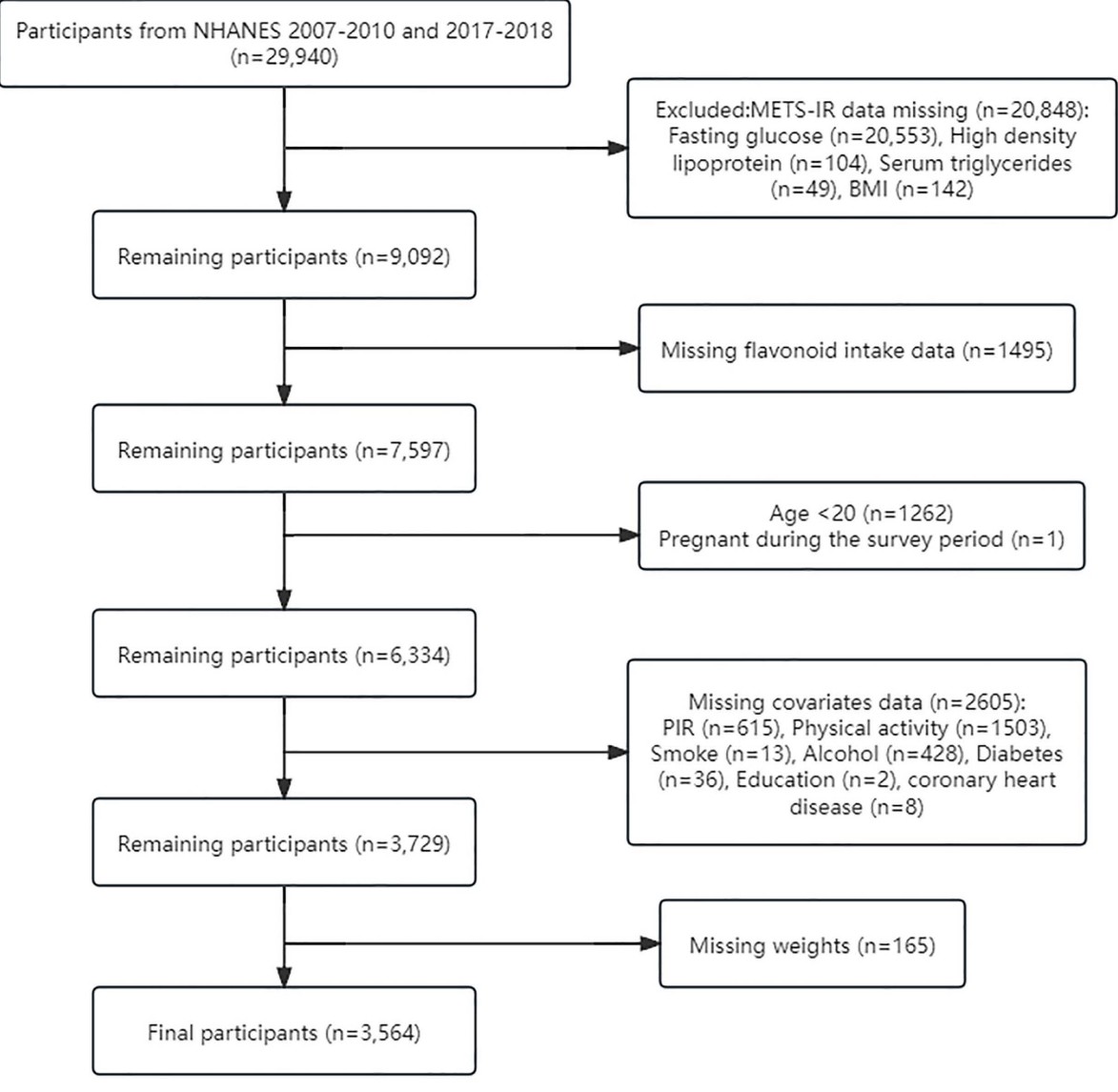

**Fig 1. Flow chart of study participants.**

## 2.4. Assessment of covariates

To assess the influence of confounding variables on the relationship between dietary flavonoid intake and METS-IR, the study incorporated the following covariates: Demographic variables such as age, sex, race, education, and PIR; lifestyle factors such as alcohol consumption, smoking status, energy intake, and physical activity; and disease factors such as coronary heart disease, hypertension, and diabetes.

We divided participants ages into 20–39 years, 40–59 years, and ≥ 60 years. Ethnicity was categorized into Mexican Americans, non-Hispanic whites, non-Hispanic blacks, other Hispanics, and other races. Education level was classified as below high school, high school or equivalent, partial university or AA degree, and university graduate or above. To assess the impact of socioeconomic status on health outcomes, we utilized the PIR-a core socioeconomic metric in NHANES

that quantifies income relative to household needs. PIR is calculated as the ratio of total family income to the federally established poverty threshold [27]. Consistent with U.S. Census Bureau poverty guidelines and established NHANES literature [28–30], we classified PIR into three tiers: low (PIR < 1.3), medium (1.3–3.5), and high (PIR > 3.5). This classification aligns with federal assistance eligibility criteria while optimizing capture of socioeconomic gradients in metabolic disease research. Smoking status was divided into three groups: Never (smoked less than 100 cigarettes in life), former (smoked more than 100 cigarettes in life and do not smoke at all currently) and current smokers (smoked more than 100 cigarettes in life and smoke some days or daily). Alcohol intake was categorized as never (had < 12 drinks in lifetime), former (had ≥ 12 drinks in 1 year and did not drink last year, or did not drink last year but drank ≥ 12 drinks in lifetime), mild (defined as two drinks a day for men and one for women), moderate (three drinks a day for men and two drinks a day for women, or binge drinking on 2–4 days per day) and heavy (defined as at least four drinks per day for men, at least three drinks per day for women, or binge drinking on at least 5 days per day) [31]. Physical activity (PA) was divided into two levels: low level, defined as less than 599 metabolic equivalent (MET) per week; High levels, defined as 599 or more MET per week [32]. When asked "Have you ever been told you have coronary artery disease?" When the answer to the question is "yes," they are diagnosed with coronary heart disease. Based on questionnaire and physical examination results, participants were diagnosed with hypertension if they satisfied one of three criteria: (1) mean systolic blood pressure ≥ 140 mmHg or mean diastolic blood pressure ≥ 90 mmHg; (2) Taking a prescription for hypertension; (3) Hypertension diagnosed by a doctor or health professional. Diabetes is diagnosed if any of the following conditions are met: (1) doctor reported diabetes diagnosis, (2) fasting glucose (mmol/l) ≥ 7.0, (3) use of diabetes medication or insulin [33].

## 2.5. Statistical analysis

Given that NHANES employed a complex stratified, multi-stage sampling design, we weighted the data according to the NHANES guidelines in order to obtain statistical outcomes representative of the overall population. Categorical variables were analyzed using Chi-square tests and expressed as proportions (n) and percentages (%) when baseline characteristics were analyzed for participants; If a continuous variable conformed to a normal distribution, it was expressed as mean ± standard deviation (SD) and a t-test was used. If it exhibited skewed distribution, it was expressed as median (25th, 75th percentiles) [M(P25, P75)], and the Shapiro-Wilk test was utilized.

We used a weighted generalized linear regression model to assess the relationship between flavonoid intake and the METS-IR, with the very low concentration group as a reference. The crude model is untuned. Model 1 was adjusted for sex, age, and race. Model 2 incorporated additional adjustments beyond Model 1 for education, PIR, smoking status, drinking status, calorie intake, PA, hypertension, and coronary atherosclerosis. Weighted restricted cubic splines (RCS) and smooth curve fitting were used to assess potential non-linear associations between flavonoid intake and IR, with threshold effects analyzed using log-likelihood ratio testing. In addition, we performed subgroup analyses and interaction assessments for variables such as sex, age, race, education level, PIR, smoking status, drinking status, PA, BMI, hypertension, coronary atherosclerosis, and diabetes. All statistical analyses were performed using R software (version 4.4.1), R packages "RNHANES", "rms" and "survey". Using bilateral tests, P < 0.05 was considered statistically significant.

## 2.6. Network pharmacology and molecular docking

### 2.6.1. Acquisition of differential genes of IR-IS.
RNA sequencing data of IR and insulin-sensitive (IS) adipose tissue samples in the GSE20950 and GSE26637 datasets were downloaded from NCBI Gene Expression Omnibus (GEO) (https://www.ncbi.nlm.nih.gov/geo/). The Limma package in R software (version 4.3.1) was used to study the differential expression of mRNA. We defined "Adjusted P <0.05 and log1.3 (fold change) > 1 or log1.3 (fold change) <-1"as the threshold for screening differential mRNA expression. To further explore potential target functions, we performed Gene Ontology (GO) and Kyoto Encyclopedia of Genes and Genomes (KEGG) functional enrichment analysis on the data.

**2.6.2. Screening genes related to IR.** Weighted correlation network analysis (WGCNA) was used to screen IR-related genes. First, we calculated the Median Absolute Deviation (MAD) of each gene using gene expression profiles, excluded the top 50% of the genes with the smallest MAD, and removed outlier genes and samples using the goodSamplesGenes method of the R software package WGCNA. WGCNA is further used to construct a scale-free co-expression network. sensitivity was set to 3. To further analyze modules, we calculated dissimilarity of module eigen genes, chose a cut line for module dendrogram and merged modules with distances less than 0.25. The intersection of the obtained genes and IR-IS differential genes represented differential genes associated with IR.

**2.6.3. Dietary flavonoid-IR related target acquisition.** According to the Database of Flavonoid Values for USDA Food Codes, 29 dietary flavonoids were selected for further study. TCMSP database (https://old.tcmsp-e.com/tcmsp.php) was used to analyze 29 dietary flavonoid targets. Simultaneously, from PubChem, 29 kinds of dietary flavonoid chemical structures were obtained and imported into the SwissTargetPrecision database (http://www.swisstargetprediction.ch/), with a threshold (probability > 0.6) set to obtain a possible target for each dietary flavonoid. Compounds with missing targets were excluded. Venny R software package was used to draw the Venn diagram of dietary flavonoid targets and IR-related genes. Intersection target represented Dietary flavonoid-IR related target. We quantified occurrence frequencies of all intersecting targets to measure target range within dietary flavonoid.

**2.6.4. PPI and gene-gene correlation.** To comprehensively understand genes and their proteins relationships, we conducted Protein-Protein Interaction (PPI) and gene-gene correlation analysis of dietary flavonoid-IR related targets. The STRING database (https://string-db.org/) was used to analyze the obtained dietary flavonoid-IR related targets for PPI, with the species "Homo sapiens" elected to generate the PPI networks. Minimum required interaction score was set to 0.4. To explore gene-gene correlations, the R software pheatmap package was used to visualize the correlations displayed by dietary flavonoid-IR related targets in GSE20950 and GSE26637 datasets. Hub genes were screened based on PPI and gene-gene correlation results, and the expression levels of Hub genes in insulin resistance and insulin sensitivity patients were visualized. Subsequently, comprehensively understand the Hub genes were enriched by KEGG and GO.

**2.6.5. Molecular docking of dietary flavonoid with hub genes.** To evaluate the dietary flavonoid binding affinity of key genes, the molecular docking method was used for analysis. CB - Dock2 (https://cadd.labshare.cn/cb-dock2/php/index.php) was used to simulate dietary flavonoid and the hub genes, with Vina score combinations employed to evaluate ligand-target affinity combinations. It is commonly accepted that a score < −5.0 kcal/mol indicates a stronger binding interaction between the two entities.

Subsequently, a weighted scoring system was employed to evaluate the interaction strength between flavonoids and potential targets. Initially, the Vina binding energy (kcal/mol) obtained from molecular docking was linearly transformed and normalized to a 0–1 scale (with 1 representing the strongest binding affinity). Subsequently, an integrated scoring model was constructed by incorporating three key parameters: 1) the sum of normalized binding energies for each target; 2) the number of flavonoid compounds capable of binding to each target (target frequency); and 3) the coverage of flavonoid subclasses (category breadth). The final weighted score was calculated as the product of these three parameters to quantitatively assess the relative importance of each target within the flavonoid interaction network. This multidimensional integration approach overcomes the limitations of relying solely on binding energy or occurrence frequency, enabling a more comprehensive identification of key targets.

## 2.7. Effect of dietary flavonoid on IR of 3T3-L1 adipocytes

3T3-L1 preadipocytes (Shanghai Fuheng Biological Co., LTD) were seeded into 12-well plates at a density of $5 \times 10^4$ cells per well and differentiated into mature adipocytes as previously described [34]. To establish an insulin resistance (IR) model, mature adipocytes were treated with 300 μmol/L palmitic acid for 6 hours [35]. The IR cells were then supplemented with cyanidin, chosen as a representative anthocyanidin due to its prevalence in nature [35], or with hesperidin.

Cells were cultured in complete medium containing 0, 50, or 100 nmol/L insulin (CSP001–10, Shanghai Zhong Qiao Xin Zhou Biotechnology) for 24 hours.

Mitochondrial membrane potential was assessed using the JC-1 assay kit (C2006, Beyotime), and glucose uptake was measured using the Glucose Uptake Fluorometric Assay Kit (E-BC-F041, Elabscience), both according to the manufacturers' protocols.

For gene expression analysis, total RNA was extracted from cells using TRIzol reagent. A total of 1 μg RNA was reverse-transcribed into cDNA. Quantitative real-time PCR (RT-qPCR) was performed using SYBR Green Master Mix, with 100 ng of cDNA used as template per reaction. The mRNA expression level of PIK3 CG was normalized to β-actin and calculated via the $2^{-\Delta\Delta Ct}$ method.All cell culture data are presented as mean ± standard deviation from at least three independent experiments. Statistical significance was determined by one-way analysis of variance (ANOVA) followed by Tukey's post-hoc test, using a p-value < 0.05 as the threshold for significance.

## 3. Results

### 3.1. Baseline characteristics

Our study encompassed 3,564 participants, comprising 1,824 (51.52%) males and 1,740 (48.48%) females. The average METS-IR score for all participants was 42.17 ± 0.28. The median intake of Isoflavones, Anthocyanidins, total Flavan-3-ols, Flavanones, Flavones, Flavonols and total flavonols for all subjects was 0.02 mg/d, 2.56 mg/d, 19.18 mg/d, 0.64 mg/d, 0.63 mg/d, 14.71 mg/d and 78.39 mg/d, respectively. According to the concentration gradient of total flavonide intake, referring to its distribution characteristics and nutritional thresholds, classifications were: Very low [0,50]mg/d, Low (50,200]mg/d, Medium (200,500]mg/d and High >500mg/d, with median values of 22.48 mg/d, 95.98 mg/d, 324.66 mg/d and 791.59 mg/d respectively. Findings revealed that compared with the extremely low flavonoid intake group, individuals with higher flavonoid intake were more likely to be aged 40–59, non-Hispanic white, possess higher PIR, achieve superior education levels, drink less alcohol, never smoke, lack hypertension, maintain higher total calorie intake and have a higher fasting glucose level. No significant differences existed between groups regarding gender, BMI, PA, hypertension, coronary heart disease (CHD), diabetes, HDL, triglycerides(TG), and fasting insulin (FINS) levels (Table 1).

### 3.2. Associations between dietary flavonoid intake and METS-IR

We employed weighted generalized linear regression model to assess the relationships between flavonoid intake and METS-IR. In Model 2, we comprehensively adjusted for age, sex, race, education, PIR, smoking status, drinking status, calorie intake, physical activity, hypertension, and coronary atherosclerosis. Compared with the very low concentration group, Anthocyanidins (P < 0.0001) and Flavanones (P < 0.001) in the high concentration group were significantly negatively correlated with METS-IR. Notably, compared with the very low concentration group, the low concentration group (β = −1.39, P = 0.02) and the moderate concentration group (β = −2.84, P = 0.001) of total flavonol intake had the potential to reduce METS-IR. Trend test results showed that Anthocyanidins (P for trend<0.0001) and Flavanones (P for trend<0.001) had a linear trend with METS-IR (Table 2).

### 3.3. The non-linear relationships between flavonoids intake and METS-IR

We used an RCS to explore the non-linear relationship between flavonoid intake and METS-IR. The results showed that total flavonoids (P for non-linearity = 0.0115), Anthocyanidins (P for non-linearity = $2.58 \times 10^{-9}$), Isoflavones (P for non-linearity = 0.0009), Flavan-3-ols (P for non-linearity = 0.0027), Flavanones (P for non-linearity = 0.0008), Flavones (P for non-linearity = 0.0482) and Flavonols (P for non-linearity = 0.0109) had statistically significant nonlinear correlations with METS-IR. Correlations between Isoflavones, Flavones, Flavan-3-ols and METS-IR displayed J-shaped curve, while correlations between Anthocyanidins, Flavanones, Flavonols and total flavones and METS-IR is U-shaped curve. We

**Table 1. Baseline characteristic table of participants grouped according to total dietary flavonoid intake.**

| Variable | Total (n = 3564) | Very low (n = 1458) | Low (n = 1115) | Moderate (n = 568) | High (n = 423) | P-value |
|---|---|---|---|---|---|---|
| **Age/years, n (%)** | | | | | | < 0.0001 |
| 20-39 | 1233(38.52) | 596(46.05) | 353(34.25) | 181(39.11) | 103(26.51) | |
| 40-59 | 1275(39.59) | 470(35.45) | 421(40.28) | 198(37.74) | 186(51.52) | |
| ≥60 | 1056(21.89) | 392(18.50) | 341(25.47) | 189(23.15) | 134(21.96) | |
| **Sex, n (%)** | | | | | | 0.73 |
| Female | 1740(48.48) | 713(49.97) | 541(46.89) | 281(47.56) | 205(48.88) | |
| Male | 1824(51.52) | 745(50.03) | 574(53.11) | 287(52.44) | 218(51.12) | |
| **Race, n (%)** | | | | | | < 0.0001 |
| Mexican American | 545(7.24) | 257(8.72) | 196(8.43) | 68(5.50) | 24(2.51) | |
| Non-Hispanic Black | 623(9.37) | 280(10.55) | 194(9.85) | 96(8.80) | 53(5.69) | |
| Non-Hispanic White | 1777(72.73) | 706(71.87) | 508(69.29) | 286(73.61) | 277(81.66) | |
| Other Hispanic | 336(4.23) | 137(4.15) | 127(5.40) | 55(4.22) | 17(1.89) | |
| Other Race | 283(6.44) | 78(4.71) | 90(7.04) | 63(7.88) | 52(8.24) | |
| **PIR, n (%)** | | | | | | 0.001 |
| <1.3 | 940(16.77) | 466(21.35) | 274(14.39) | 121(15.10) | 79(11.28) | |
| 1.3-3.5 | 1351(34.75) | 573(36.40) | 413(32.81) | 216(36.86) | 149(32.10) | |
| >3.5 | 1273(48.48) | 419(42.25) | 428(52.80) | 231(48.04) | 195(56.62) | |
| **Education, n (%)** | | | | | | < 0.001 |
| Less than high school | 701(12.00) | 346(14.36) | 218(11.26) | 85(9.84) | 52(9.54) | |
| High school or equivalent | 811(23.25) | 383(26.91) | 218(19.64) | 113(20.75) | 97(23.91) | |
| Some college or AA degree | 1102(30.85) | 454(32.41) | 340(31.23) | 172(28.98) | 136(27.81) | |
| College graduate or above | 950(33.90) | 275(26.32) | 339(37.87) | 198(40.43) | 138(38.74) | |
| **BMI, n (%)** | | | | | | 0.14 |
| <30 | 2268(65.40) | 891(63.33) | 712(65.42) | 385(70.79) | 280(65.02) | |
| ≥30 | 1296(34.60) | 567(36.67) | 403(34.58) | 183(29.21) | 143(34.98) | |
| **Drinking status, n (%)** | | | | | | < 0.001 |
| Former | 434(9.24) | 211(10.74) | 116(7.80) | 60(7.89) | 47(9.80) | |
| Heavy | 770(21.88) | 377(27.45) | 223(19.64) | 96(17.35) | 74(16.60) | |
| Mild | 1391(42.98) | 489(36.87) | 458(45.88) | 244(47.43) | 200(48.43) | |
| Moderate | 599(17.86) | 240(17.47) | 190(18.04) | 109(19.80) | 60(16.36) | |
| Never | 370(8.04) | 141(7.48) | 128(8.64) | 59(7.54) | 42(8.81) | |
| **Smoking status, n (%)** | | | | | | < 0.001 |
| Now | 675(17.36) | 358(22.49) | 162(13.12) | 86(15.71) | 69(14.37) | |
| Former | 913(26.42) | 365(25.61) | 283(26.84) | 146(28.14) | 119(25.79) | |
| Never | 1976(56.22) | 735(51.90) | 670(60.04) | 336(56.15) | 235(59.84) | |
| **PA, n (%)** | | | | | | 0.38 |
| High | 2919(83.08) | 1186(82.15) | 929(84.42) | 466(84.58) | 338(80.97) | |
| Low | 645(16.92) | 272(17.85) | 186(15.58) | 102(15.42) | 85(19.03) | |
| **CHD, n (%)** | | | | | | 0.64 |
| No | 3428(96.90) | 1400(97.45) | 1078(96.45) | 545(97.08) | 405(96.19) | |
| Yes | 136(3.10) | 58(2.55) | 37(3.55) | 23(2.92) | 18(3.81) | |
| **Hypertension, n (%)** | | | | | | 0.01 |
| No | 2192(66.47) | 895(66.09) | 670(65.07) | 380(74.59) | 247(61.43) | |
| Yes | 1372(33.53) | 563(33.91) | 445(34.93) | 188(25.41) | 176(38.57) | |
| **Diabetes, n (%)** | | | | | | 0.06 |
| No | 3165(91.76) | 1281(91.76) | 996(91.09) | 516(94.85) | 372(89.79) | |

*(Continued)*

**Table 1.** (Continued)

| Variable | Total (n = 3564) | Very low (n = 1458) | Low (n = 1115) | Moderate (n = 568) | High (n = 423) | P-value |
|---|---|---|---|---|---|---|
| Yes | 399(8.24) | 177(8.24) | 119(8.91) | 52(5.15) | 51(10.21) | |
| Total calories (kcal/d) | 2161.24 ± 15.35 | 2019.03 ± 23.11 | 2249.82 ± 29.03 | 2264.83 ± 48.20 | 2241.96 ± 46.11 | < 0.0001 |
| HDL (mg/dL) | 54.36 ± 0.46 | 53.59 ± 0.61 | 54.76 ± 0.75 | 55.41 ± 0.85 | 54.46 ± 0.96 | 0.35 |
| TG (mg/dL) | 124.68 ± 2.25 | 125.54 ± 2.35 | 124.05 ± 3.88 | 121.01 ± 5.25 | 127.80 ± 6.19 | 0.78 |
| Fasting glucose (mg/dL) | 105.50 ± 0.70 | 105.37 ± 0.99 | 106.55 ± 1.36 | 102.61 ± 0.91 | 106.79 ± 1.57 | 0.04 |
| FINS (pmol/L) | 12.01 ± 0.29 | 12.19 ± 0.35 | 12.13 ± 0.39 | 11.00 ± 0.53 | 12.42 ± 0.83 | 0.07 |
| METS-IR | 42.17 ± 0.28 | 43.49 ± 0.44 | 41.65 ± 0.42 | 39.47 ± 0.72 | 42.74 ± 0.75 | < 0.0001 |
| Total Isoflavones (mg/d) | 0.02 (0.00,0.14) | 0.01 (0.00,0.06) | 0.02 (0.00,0.84) | 0.02 (0.00,0.16) | 0.01 (0.00,0.20) | < 0.0001 |
| Total Anthocyanidins (mg/d) | 2.56 (0.11,15.26) | 0.59 (0.00, 3.60) | 9.90 (1.69,34.64) | 5.51 (0.39,23.83) | 3.64 (0.16,18.15) | < 0.0001 |
| Total Flavan-3-ols (mg/d) | 19.18 (5.75,211.19) | 4.97 (1.98, 10.25) | 22.73 (11.26, 47.10) | 267.41 (198.18, 339.51) | 717.98 (532.28,1012.58) | < 0.0001 |
| Total Flavanones (mg/d) | 0.64 (0.09,14.88) | 0.18 (0.00, 1.22) | 10.41 (0.35,35.29) | 1.77 (0.18,23.17) | 0.50 (0.12,15.81) | < 0.0001 |
| Total Flavones (mg/d) | 0.63 (0.24,1.22) | 0.32 (0.11,0.73) | 0.76 (0.34,1.38) | 0.88 (0.43,1.62) | 0.98 (0.53,1.87) | < 0.0001 |
| Total Flavonols (mg/d) | 14.71 (8.23,25.24) | 7.91 (4.83,12.36) | 14.62 (10.09,20.75) | 23.14 (17.58,28.70) | 41.68 (31.95,54.21) | < 0.0001 |
| Total Sum of all 29 flavonoids (mg/d) | 78.39 (28.67,271.42) | 22.48 (12.53, 34.01) | 95.98 (68.78, 138.83) | 324.66 (254.15, 403.24) | 791.59 (595.73,1075.31) | < 0.0001 |

Very low: [0,50]mg/d, Low: (50,200]mg/d, Moderate: (200,500]mg/d, High: > 500mg/d

BMI, body mass index, PIR, family poverty income ratio, PA, physical activity, CHD, coronary heart disease, HDL, high density lipoprotein, TG, serum triglycerides, FINS, fasting insulin.

observed a negative association with METS-IR when total flavone intake remained below 337.12 mg/d. The METS-IR values were lowest when Anthocyanidins, Flavan-3-ols and Flavanones were at 35.34 mg/d, 33.41 mg/d and 29.85 mg/d. METS-IR decreases gradually with increasing intake at less than 25.78 mg/d in Flavonols. Notably, METS-IR continued to decline with increased intake of Isoflavones and Flavones, although the decline decelerated slightly (Fig 2).

### 3.4 Subgroup analysis and interaction tests

Subgroup analysis and interaction tests showed that age, sex, race, PIR, education, smoking status, drinking status, physical activity, BMI, hypertension, coronary heart disease, and diabetes did not substantially influence the association between total flavone intake and METS-IR (P for interaction>0.05) (Table 3). A substantial negative correlation between Anthocyanidins and Flavanones and METS-IR. Therefore, we performed an additional subgroup analysis on them. The results showed that when stratified by sex (P for interaction = 0.01) and coronary heart disease (P for interaction = 0.01), significant interactions existed between Anthocyanidins and METS-IR (S1 Table.). In Flavanones subgroup analysis, we observed no significant interactions among alternative variables except gender (P for interaction = 0.003) (S2 Table.). Meanwhile, in flavonoid sub-class interaction testing, we discovered that except for Flavones which potentially influenced the relationship between total flavonoid intake and IR (P for interaction = 0.04), no significant interactions existed among other subclasses (S3 Table.).

### 3.5. Obtain differential genes related to IR

Following the aforementioned methods, 821 IR-IS differential genes were identified, comprising 528 up-regulated and 293 down-regulated genes (Figs 3A and 3B). As illustrated in Figs 3C and 3D, GO and KEGG enrichment analysis revealed

**Table 2. Associations between dietary flavonoid intake and METS-IR.**

| Exposure | Very low | Low | | Moderate | | High | | P for trend |
|---|---|---|---|---|---|---|---|---|
| | | β (95% CI) | P-value | β (95% CI) | P-value | β (95% CI) | P-value | |
| **Total Sum of all 29 flavonoids (mg/d)** | | | | | | | | |
| Crude model | Reference | −1.84(−3.05,-0.63) | 0.004 | −4.03(−5.63,-2.42) | <0.0001 | −0.75(−2.57, 1.06) | 0.41 | 0.01 |
| Model 1 | Reference | −2.06(−3.21,-0.90) | <0.001 | −4(−5.61,-2.39) | <0.0001 | −0.62(−2.40, 1.16) | 0.49 | 0.02 |
| Model 2 | Reference | −1.39(−2.54,-0.25) | 0.02 | −2.84(−4.42,-1.27) | 0.001 | −0.29(−2.04, 1.45) | 0.73 | 0.12 |
| **Total Isoflavones (mg/d)** | | | | | | | | |
| Crude model | Reference | −0.27(−1.59, 1.04) | 0.68 | 0.3(−0.92, 1.52) | 0.62 | −1.7(−3.03,-0.38) | 0.01 | 0.02 |
| Model 1 | Reference | −0.23(−1.57, 1.10) | 0.72 | 0.11(−1.06, 1.27) | 0.85 | −1.66(−3.04,-0.27) | 0.02 | 0.03 |
| Model 2 | Reference | 0.1(−1.20, 1.39) | 0.88 | 0.52(−0.73, 1.77) | 0.40 | −0.69(−1.94, 0.56) | 0.26 | 0.39 |
| **Total Anthocyanidins (mg/d)** | | | | | | | | |
| Crude model | Reference | −1.78(−3.21,-0.34) | 0.02 | −3.08(−4.39,-1.77) | <0.0001 | −5.03(−6.44,-3.62) | <0.0001 | <0.0001 |
| Model 1 | Reference | −2(−3.48,-0.51) | 0.01 | −3.12(−4.49,-1.75) | <0.0001 | −5(−6.43,-3.56) | <0.0001 | <0.0001 |
| Model 2 | Reference | −1.61(−2.98,-0.24) | 0.02 | −2.59(−3.80,-1.38) | <0.001 | −3.69(−5.18,-2.19) | <0.0001 | <0.0001 |
| **Total Flavan-3-ols (mg/d)** | | | | | | | | |
| Crude model | Reference | −2.53(−3.79,-1.27) | <0.001 | −2.67(−4.14,-1.19) | <0.001 | −2.16(−3.60,-0.72) | 0.004 | 0.01 |
| Model 1 | Reference | −2.74(−3.93,-1.56) | <0.0001 | −2.72(−4.10,-1.33) | <0.001 | −2.06(−3.46,-0.66) | 0.01 | 0.02 |
| Model 2 | Reference | −2.19(−3.34,-1.04) | <0.001 | −1.7(−2.88,-0.52) | 0.01 | −1.27(−2.61, 0.08) | 0.06 | 0.1 |
| **Total Flavanones (mg/d)** | | | | | | | | |
| Crude model | Reference | −2.21(−3.67,-0.75) | 0.004 | −4.71(−6.39,-3.04) | <0.0001 | −3.63(−5.16,-2.10) | <0.0001 | <0.0001 |
| Model 1 | Reference | −2.09(−3.61,-0.57) | 0.01 | −4.61(−6.37,-2.86) | <0.0001 | −4.01(−5.56,-2.46) | <0.0001 | <0.0001 |
| Model 2 | Reference | −1.58(−2.99,-0.17) | 0.03 | −3.49(−5.26,-1.72) | <0.001 | −2.99(−4.62,-1.36) | <0.001 | <0.001 |
| **Total Flavones (mg/d)** | | | | | | | | |
| Crude model | Reference | −0.9(−2.46, 0.65) | 0.25 | −1.65(−3.03,-0.27) | 0.02 | −2.29(−3.72,-0.85) | 0.003 | 0.001 |
| Model 1 | Reference | −1.06(−2.60, 0.48) | 0.17 | −1.59(−2.98,-0.21) | 0.03 | −2.39(−3.88,-0.91) | 0.002 | 0.001 |
| Model 2 | Reference | −0.44(−1.74, 0.85) | 0.49 | −0.6(−2.19, 0.99) | 0.44 | −1.21(−2.45, 0.03) | 0.06 | 0.1 |
| **Total Flavonols (mg/d)** | | | | | | | | |
| Crude model | Reference | −1.51(−2.97,-0.04) | 0.04 | −2.61(−4.18,-1.04) | 0.002 | −1.96(−3.68,-0.25) | 0.03 | 0.02 |
| Model 1 | Reference | −1.69(−3.17,-0.20) | 0.03 | −2.8(−4.28,-1.33) | <0.001 | −2.32(−3.97,-0.68) | 0.01 | 0.004 |
| Model 2 | Reference | −1.29(−2.76, 0.18) | 0.08 | −2.13(−3.69,-0.57) | 0.01 | −1.86(−3.60,0.01) | 0.05 | 0.04 |

Crude model: no covariates were adjusted. Model 1: adjusted by age, sex, and race. Model 2: adjusted by age, sex, race, education, PIR, smoking status, drinking status, caloric intake, physical activity, coronary heart disease, and hypertension.

Total Sum of all 29 flavonoids: [0,50]mg/d, (50,200]mg/d, (200,500]mg/d, >500mg/d

Total Anthocyanidins: [0,0.15]mg/d, (0.15,2.36]mg/d, (2.36,13.6]mg/d, >13.6mg/d

Total Isoflavones: [0,0]mg/d, (0.001,0.015]mg/d, (0.02,0.1]mg/d, >0.1mg/d

Total Flavan_3_ols: [0,5.3]mg/d, (5.03,17]mg/d, (17,170] mg/d, >170mg/d

Total Flavanones: [0,0.1]mg/d, (0.1,1]mg/d, (1,20]mg/d, >20mg/d

Total Flavones: [0,0.2]mg/d, (0.2,0.6]mg/d, (0.6,1.2]mg/d, >1.2mg/d

Total Flavonols: [0,8]mg/d, (8,14]mg/d, (14,24]mg/d, >24mg/d

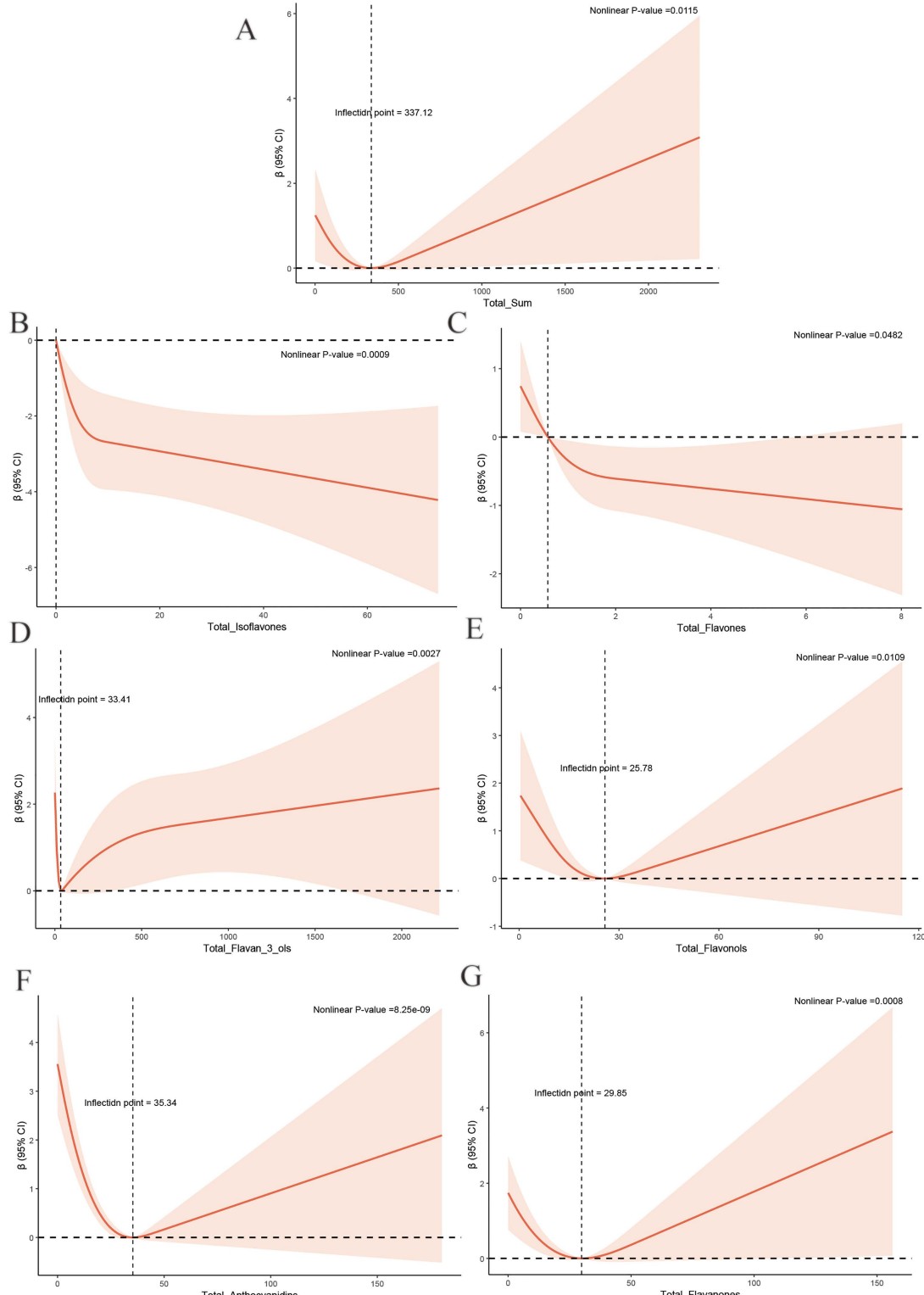

**Fig 2. The association of flavonoid intake with METS-IR by restricted cubic splines.** Models by restricted cubic splines were adjusted for sex, age, race, education, PIR, smoking status, drinking status, calorie intake, physical activity, hypertension, and coronary atherosclerosis.

**Table 3.** Subgroup analysis between total flavonoid intake and METS-IR.

| Character | Very low | Low | | Moderate | | High | | P for interaction |
|---|---|---|---|---|---|---|---|---|
| | | β (95% CI) | P-value | β (95% CI) | P-value | β (95% CI) | P-value | |
| **Age/years** | | | | | | | | 0.39 |
| 20-39 | ref | −1.58(−3.98, 0.81) | 0.19 | −2.15(−4.70, 0.40) | 0.09 | −2.11(−5.29, 1.07) | 0.18 | |
| 40-59 | ref | −2.07(−5.32, 1.17) | 0.20 | −1.87(−4.41, 0.67) | 0.14 | 0.11(−2.59, 2.82) | 0.93 | |
| ≥60 | ref | −1.98(−4.64, 0.67) | 0.14 | −1.70(−4.95, 1.55) | 0.29 | −2.62(−5.70, 0.46) | 0.09 | |
| **Sex** | | | | | | | | 0.27 |
| Female | ref | −2.61(−5.17,-0.06) | 0.05 | −1.80(−4.47, 0.87) | 0.18 | −1.87(−4.34, 0.61) | 0.13 | |
| Male | ref | −1.15(−3.21, 0.90) | 0.26 | −1.90(−3.61,-0.19) | 0.03 | −0.73(−2.80, 1.34) | 0.48 | |
| **Race** | | | | | | | | 0.33 |
| Mexican American | ref | −0.97(−5.75,3.80) | 0.66 | 1.32(−3.25,5.89) | 0.54 | −3.23(−9.41,2.95) | 0.27 | |
| Non-Hispanic White | ref | −2.40(−4.78,-0.03) | 0.05 | −2.49(−4.19,-0.79) | 0.01 | −1.82(−3.90, 0.25) | 0.08 | |
| Non-Hispanic Black | ref | −2.56(−6.43, 1.30) | 0.17 | −3.07(−8.03, 1.88) | 0.20 | 2.57(−2.35, 7.49) | 0.27 | |
| Other Hispanic | ref | 0.27(−3.91,4.44) | 0.89 | 1.62(−2.50,5.74) | 0.41 | −1.80(−7.57,3.98) | 0.51 | |
| Other Race | ref | −3.55(−9.88, 2.78) | 0.21 | −4.65(−13.07, 3.77) | 0.21 | −3.5(−11.51, 4.52) | 0.31 | |
| **PIR** | | | | | | | | 0.51 |
| <1.3 | ref | −3.37(−6.42,-0.33) | 0.03 | −3.17(−6.20,-0.13) | 0.04 | 0.22(−2.96, 3.41) | 0.89 | |
| 1.3-3.5 | ref | −1.52(−4.62, 1.58) | 0.32 | −0.89(−3.60, 1.83) | 0.51 | −0.63(−3.43, 2.18) | 0.65 | |
| >3.5 | ref | −1.58(−3.95, 0.79) | 0.18 | −2.49(−5.22, 0.24) | 0.07 | −2.39(−5.11, 0.32) | 0.08 | |
| **Education** | | | | | | | | 0.22 |
| Less than high school | ref | −2.17(−5.13, 0.80) | 0.14 | −0.59(−2.98, 1.80) | 0.61 | 0.50(−2.98, 3.99) | 0.77 | |
| High school or equivalent | ref | −0.60(−4.42, 3.21) | 0.75 | −1.96(−5.14, 1.23) | 0.22 | −0.11(−3.34, 3.12) | 0.94 | |
| Some college or AA degree | ref | −2.25(−4.21,-0.30) | 0.03 | −1.92(−4.29, 0.45) | 0.11 | −0.52(−2.87, 1.82) | 0.65 | |
| College graduate or above | ref | −3.06(−6.04,-0.07) | 0.05 | −3.35(−6.62,-0.08) | 0.05 | −4.09(−6.79,-1.39) | 0.005 | |
| **Coronary heart disease** | | | | | | | | 0.81 |
| No | ref | −1.96(−3.90,-0.03) | 0.05 | −1.92(−3.53,-0.32) | 0.02 | −1.40(−3.15, 0.36) | 0.11 | |
| Yes | ref | −6.93(−10.87,-2.98) | 0.17 | −8.17(−13.80,-2.53) | 0.21 | −6.55(−11.03,-2.07) | 0.20 | |
| **Hypertension** | | | | | | | | 0.69 |
| No | ref | −2.24(−4.15,-0.33) | 0.02 | −2.21(−4.06,-0.36) | 0.02 | −1.80(−3.77, 0.17) | 0.07 | |
| Yes | ref | −1.39(−4.46, 1.68) | 0.36 | −1.52(−4.08, 1.04) | 0.23 | −0.60(−3.25, 2.04) | 0.64 | |
| **Diabetes** | | | | | | | | 0.39 |
| No | ref | −2.22(−3.89,-0.54) | 0.01 | −2.15(−3.65,-0.66) | 0.01 | −1.80(−3.52,-0.07) | 0.04 | |
| Yes | ref | −3.66(−9.13, 1.81) | 0.18 | −3.14(−8.46, 2.18) | 0.23 | −0.51(−6.67, 5.65) | 0.86 | |
| **BMI** | | | | | | | | 0.86 |
| <30 | ref | −0.41(−1.44, 0.62) | 0.42 | −0.63(−1.47, 0.21) | 0.14 | −0.11(−1.27, 1.04) | 0.84 | |
| ≥30 | ref | −0.53(−2.88,1.82) | 0.65 | −0.33(−2.36,1.71) | 0.74 | −0.49(−2.44,1.47) | 0.61 | |
| **Drinking status** | | | | | | | | 0.44 |
| Former | ref | −1.54(−5.17, 2.09) | 0.37 | −2.04(−5.19, 1.12) | 0.18 | 0.03(−3.76, 3.82) | 0.99 | |
| Heavy | ref | −3.82(−7.37,-0.26) | 0.04 | −3.46(−6.42,-0.50) | 0.02 | −0.60(−3.50, 2.30) | 0.67 | |
| Mild | ref | −0.83(−3.76, 2.10) | 0.56 | −2.02(−4.54, 0.49) | 0.11 | −2.00(−4.64, 0.63) | 0.13 | |
| Moderate | ref | −2.13(−5.74,1.48) | 0.24 | 0.07(−4.64,4.79) | 0.98 | −0.80(−4.36,2.76) | 0.65 | |
| Never | ref | −4.91(−10.22, 0.39) | 0.07 | −4.75(−8.84,-0.67) | 0.02 | −3.59(−8.36, 1.18) | 0.13 | |
| **Smoking status** | | | | | | | | 0.26 |
| Never | ref | −2.12(−4.05,-0.19) | 0.03 | −2.66(−4.71,-0.61) | 0.01 | −2.31(−4.41,-0.21) | 0.03 | |
| Former | ref | −1.61(−5.01,1.78) | 0.34 | −0.74(−3.51,2.02) | 0.58 | −1.46(−4.39,1.46) | 0.31 | |

*(Continued)*

**Table 3.** (Continued)

| Character | Very low | Low | | Moderate | | High | | P for interaction |
|---|---|---|---|---|---|---|---|---|
| | | β (95% CI) | P-value | β (95% CI) | P-value | β (95% CI) | P-value | |
| Now | ref | −2.55(−6.48, 1.38) | 0.19 | −1.74(−5.22, 1.75) | 0.31 | 1.58(−1.48, 4.64) | 0.30 | |
| **Physical activity** | | | | | | | | 0.76 |
| High | ref | −2.30(−4.30,-0.30) | 0.03 | −1.96(−3.66,-0.27) | 0.02 | −1.61(−3.67, 0.45) | 0.12 | |
| Low | ref | −0.45(−4.01, 3.11) | 0.80 | −1.52(−5.26, 2.21) | 0.41 | −0.37(−3.58, 2.84) | 0.81 | |

Very low: [0,50]mg/d, Low: (50,200]mg/d, Moderate: (200,500]mg/d, High: > 500mg/d

Adjusted by age, sex, race, education, PIR, smoking status, drinking status, caloric intake, physical activity, coronary heart disease, and hypertension.

BMI, body mass index, PIR, family poverty income ratio.

these highly expressed genes participate in neutrophil activation involving immune, leukocyte proliferation, macrophage activation and other immune and inflammatory processes. It is related to Toll−like receptor signaling pathway, NF-κB signaling pathway and other signaling pathways. Low expression genes are enriched in fatty acid catabolic processes, carboxylic acid catabolic processes, organic acid catabolic processes and numerous other lipid metabolism-related processes. It is related to AMP-activated protein kinase (AMPK) signaling pathway, FoxO signaling pathway and other signaling pathways. WGCNA identified a total of 43 co-expression modules, among which 10 modules achieved P-value < 0.05 (Figs 4A–4C). By extracting the genes from these modules, 320 IR-related genes were obtained. The intersection of IR-related genes and IR-IS differential genes yielded 157 genes representing differential genes related to IR (Fig 4D).

### 3.6. Dietary flavonoid-IR related target acquisition

Based on NHANES database analytical results, combined with TCMSP and SwissTargetPrecision database, we obtained 12 dietary flavonoid related targets (S4 Table.). As demonstrated in Fig 5A, intersections of each dietary flavonoid target and differential genes related to IR were calculated, with intersection gene occurrence frequencies quantified. As shown in Fig 5B, 5 intersection genes were identified. PIK3 CG has the highest occurrence frequency (11 times), followed by MMP9 (7 times), SYK (5 times), and CDC37 and PRKCB (only once).

### 3.7. Hub genes of dietary flavonoid-IR related target acquisition

As shown in Fig 5C, protein interaction results of five dietary flavonoid-IR related targets demonstrated close interactions among PIK3 CG, MMP9, SYK and PRKCB, while CDC37 exhibited no interactions with any other protein. Meanwhile, gene-gene correlation (Fig 5D) showed that PIK3 CG, MMP9, SYK and PRKCB were positively correlated with each other in gene expression, while CDC37 showed no significant correlations with the other four genes. Therefore, combining PPI, gene-gene correlation results and gene occurrence frequency, we screened PIK3 CG, MMP9, SYK and PRKCB as hub genes of dietary flavonoid-IR related targets. We visualized the expression levels of hub genes in insulin-resistant and insulin-sensitive patients in the GSE20950 and GSE26637 dataset, and found that PIK3 CG, MMP9, SYK, and PRKCB were significantly overexpressed in the fat of insulin-resistant patients (Fig 5E). According to KEGG and GO enrichment analysis, it can be seen that hub genes participate in positive regulation of signal transduction, inflammatory response and other biological processes (Figs 5F and 5G, S4 Fig.). It also participates in the regulation of NF-κB signaling pathway, PI3K-Akt signaling pathway and other signaling pathways.

### 3.8. Molecular docking of hub genes with 12 dietary flavonoid species

Building upon intersection results of dietary flavonoid and hub genes, we conducted molecular docking simulations of PIK3 CG, MMP9, SYK, and PRKCB with 12 dietary flavonoid species to evaluate their interaction ability. As shown in

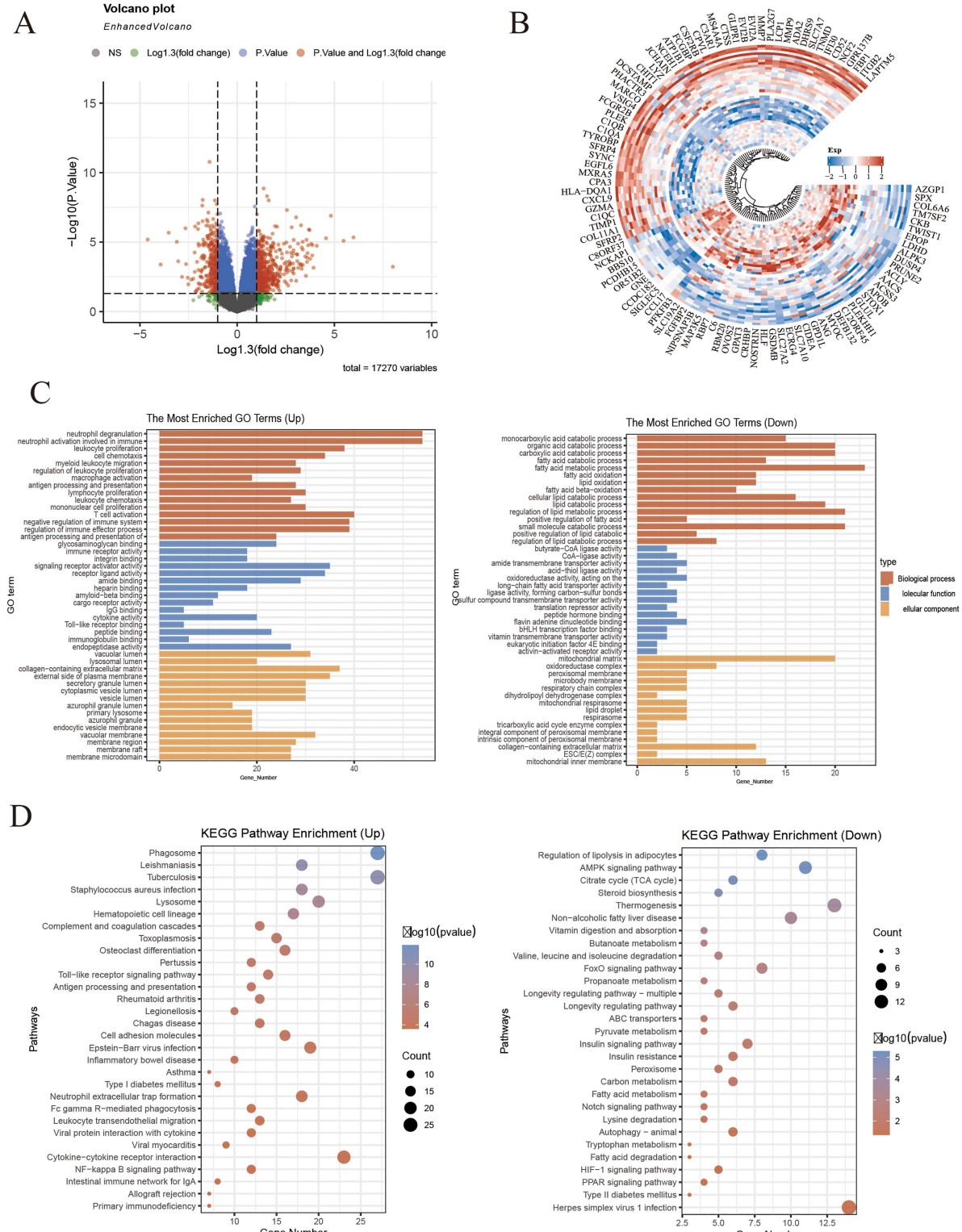

**Fig 3. Obtain differential genes related to IR. (A)** Differential genes of IR and IS adipose tissue samples from the GSE20950 and GSE26637 datasets volcano maps and **(B)** heat maps. **(C-D)** KEGG pathway enrichment results and GO term enrichment results of differentially up-regulated and differentially down-regulated genes.

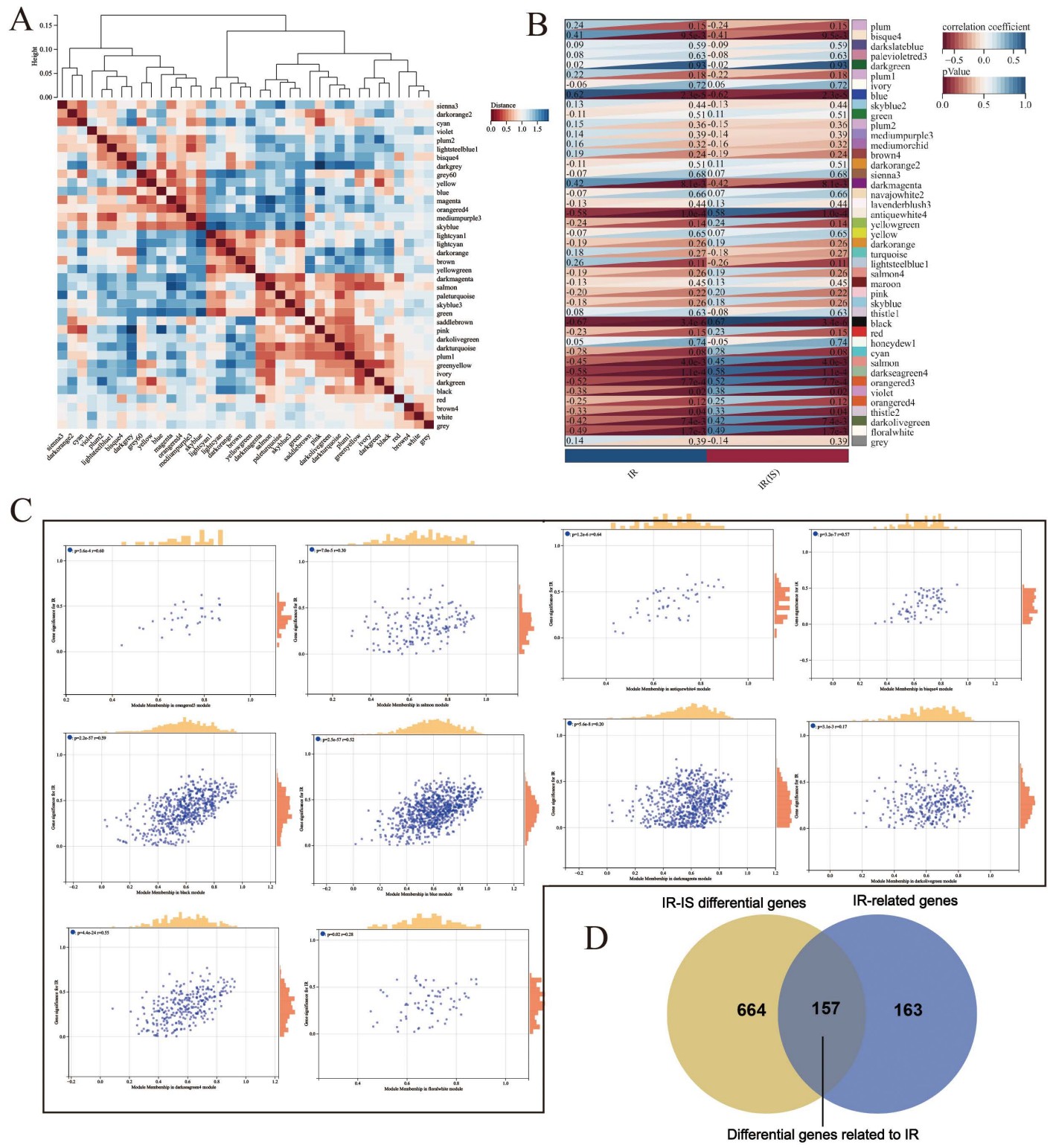

**Fig 4. Obtain differential genes related to IR. (A)** Clustering according to WGCNA's module vectors. **(B)** Heat map of correlation between modules and phenotypes. **(C)** Scatter plots of correlation between GS and MM for some modules with p < 0.05. **(D)** Venny diagram of intersection of IR-IS differential genes and IR-related genes, and the resulting genes are differential genes related to IR.

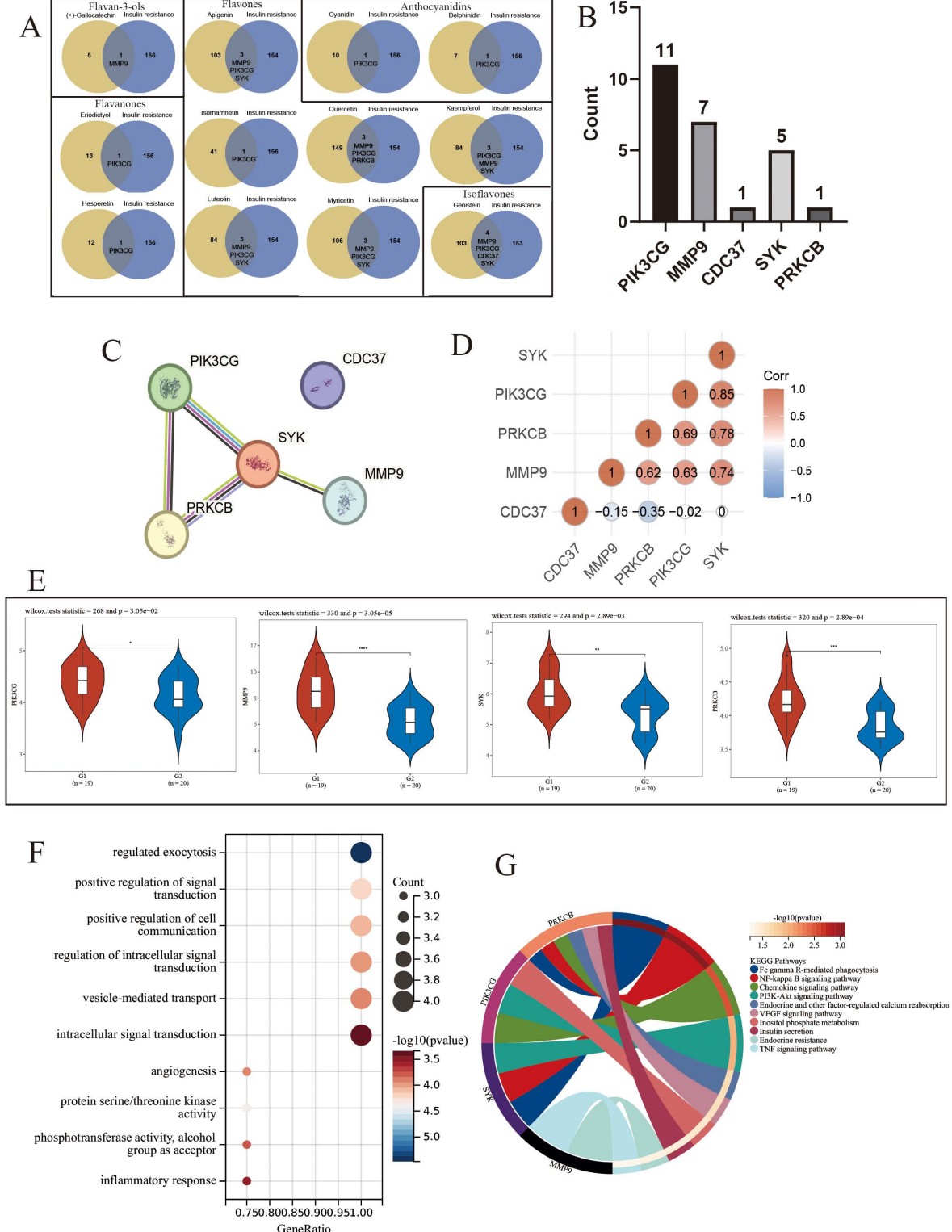

**Fig 5. Hub genes of dietary flavonoid-IR related target acquisition. (A)** Venny plot of 12 dietary flavonoid targets related to differential genes to IR. **(B)** Dietary flavonoid-IR related targets occurrence bar chart. **(C)** Dietary flavonoid-IR related targets PPI plot. **(D)** Dietary flavonoid-IR related targets gene-gene correlation heat map, both horizontal and vertical coordinates represent genes, in which different colors represent correlation coefficients,

red represents positive correlation, blue represents negative correlation, the darker the color indicates the stronger correlation. **(E)** Violin diagram of the expression of hub genes. **(F)** GO enrichment analysis of hub genes. **(G)** KEGG enrichment analysis of hub genes.

Table 4 and Figs 6A–6D, the vina score of PIK3 CG and apigenin is −8.8, that of cyanidin is −8.5, and that of delphinidin is −8.0. The vina score with eriodictyol is −8.5. Docking analyses of dietary flavonoids with the remaining 7 targets of PIK3 CG are shown in S1 Fig. As illustrated in Figs 6E and 6F, MMP9 demonstrated vina score of −9.4 with (+)-gallocatechin and −10.4 with luteolin. The results of dietary flavonoid docking with the other 5 target MMP9 are shown in S2 Fig. As depicted in Fig 6G, PRKCB and quercetin yielded a vina score of −9.3. As shown in Fig 6H, the vina score of SYK and genistein was −7.7, with docking results of dietary flavonoid against the remaining four SYK targets displayed in S2 Fig. The results of the molecular docking study of Cyanidin and Hesperetin with dietary flavonoids are shown in S3 Fig.

As presented in Table 5, the weighted analysis revealed that PIK3 CG scored significantly higher (290.4) than other targets (SYK 27.6, MMP9 26.1, PRKCB 0.41). This dominance stemmed from its broad-spectrum binding properties. Although the binding energy of individual compounds was not optimal, its systematic multi-compound and multi-subclass synergistic effects established it as the core target. While SYK and MMP9 exhibited strong binding energies, their relatively narrow action spectra resulted in significantly lower weighted scores. These findings suggest that the PI3K-Akt pathway may serve as the primary hub for flavonoids in improving insulin resistance, with SYK and MMP9 potentially playing auxiliary regulatory roles.

### 3.9. Cyanidin and hesperidin improve IR in adipocytes at an in vitro

Cross-sectional results showed that anthocyanidins and Flavanones were significantly correlated with IR, while hesperidin represented a Flavanones compound present in exceptionally elevated concentrations in citrus fruits. Consequently, we selected these two bioactive compounds as subsequent investigational targets. We used CCK-8 cell viability assay to analyze the effects of cyanidin and hesperidin on the cell viability of 3T3-L1 adipocytes (Fig 7A). To circumvent death interference with glucose uptake analysis, we chose concentrations below IC10 for subsequent studies. IR cell models received 7μmol/L cyanidin or 9 μmol/L hesperidin, respectively, and were cultured in complete culture medium containing 0, 50 and 100nmo/L insulin for 24h. JC-1 results showed that the mitochondrial membrane potential of cells increased after the addition of cyanidin and hesperidin (Fig 7B). As shown in Fig 7C, control group glucose uptake increased proportionally with insulin concentrations, while IR group glucose uptake decreased significantly compared with the control

**Table 4. List of compounds and weighted score for molecular docking.**

| Flavonoid class | Flavonoid | MMP9 | PIK3 CG | PRKCB | SYK |
|---|---|---|---|---|---|
| Anthocyanidins | Cyanidin | – | −8.5 | – | – |
| Anthocyanidins | Delphinidin | – | −8 | – | – |
| Flavan-3-ols | (+)-Gallocatechin | −9.4 | – | – | – |
| Flavanones | Eriodictyol | – | −8.5 | – | – |
| Flavanones | Hesperetin | – | −8.9 | – | – |
| Flavones | Apigenin | −9.7 | −8.8 | – | −8.1 |
| Flavones | Luteolin | −10.4 | −8.6 | – | −8.3 |
| Flavonols | Isorhamnetin | – | −8.6 | – | – |
| Flavonols | Kaempferol | −9.3 | −8.5 | – | −8.2 |
| Flavonols | Myricetin | −9.7 | −7.7 | – | −7.7 |
| Flavonols | Quercetin | −10 | −8.3 | −9.3 | – |

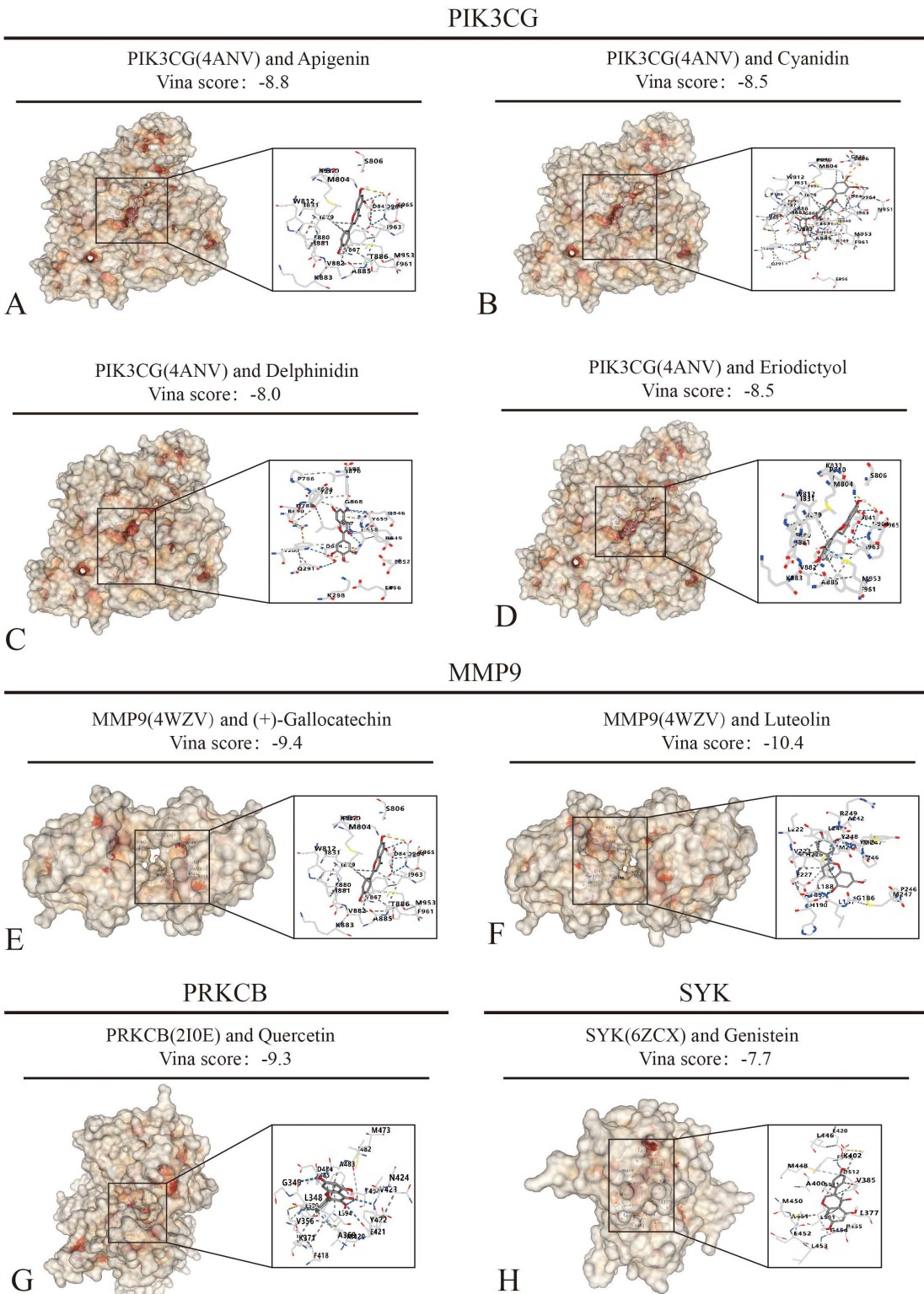

**Fig 6. Analysis of binding ability of hub genes and dietary flavonoids. (A-D)** Molecular docking of PIK3 CG with apigenin, cyandin, delphinidin and eriodictyol. **(E-F)** MMP9 is docked to the molecules of (+)-gallocatechin and luteolin. **(G)** Molecular docking of PRKCB with quercetin. **(H)** Molecular docking of SYK with genistein.

**Table 5. Weighted Scores of each target.**

| Target | Weighted Score | Rank |
|---|---|---|
| PIK3 CG | 290.4 | 1 |
| SYK | 27.6 | 2 |
| MMP9 | 26.1 | 3 |
| PRKCB | 0.41 | 4 |

group at equivalent insulin content. Following cyanidin and hesperidin administration, glucose uptake recovered significantly across varying insulin concentrations compared with IR group, reflecting improved IR. Further experimental results demonstrated that, in addition to cyanidin and hesperetin, the other three flavonoids (kaempferol, luteolin, and myricetin) also significantly reduced PIK3 CG mRNA levels (Fig 6D) and decreased TNF-α secretion in the culture supernatant (Fig 6E) at a concentration of 5μmol/L. These findings suggest that dietary flavonoids may share a common mechanism in ameliorating insulin resistance, potentially through their anti-inflammatory effects.

## 4. Discussion

This study integrated the NHANES database with network pharmacology to investigate the relationship between dietary flavonoid intake and IR, demonstrating a substantial inverse association between intake of Anthocyanidins and Flavanones and METS-IR following adjustment for baseline characteristics, lifestyle variables, and comorbidities. Furthermore, the RCS curves reveal nonlinear correlations between total flavonoids, Anthocyanidins, Isoflavones, Flavan-3-ols, Flavanones, Flavones and Flavonols and METS-IR parameters. In vitro experiments corroborated these findings, whereby anthocyanidins and hesperidin supplementation significantly restored glucose uptake in IR groups, ameliorating IR.

Recent research has shown growing scientific interest in natural flavonoids derived from dietary sources owing to their anti-inflammatory, antioxidant and improvement of lipid metabolism [36–37]. Extensive investigations have established that flavonoids possess considerable therapeutic potential for preventing and treating various metabolic diseases. A cross-sectional study by Wan et al. showed that flavonoid intake was negatively correlated with the prevalence of hyperlipidemia [38]. Catechins in green tea can significantly reduce serum total cholesterol (TC) and low density lipoprotein (LDL) cholesterol concentrations, proving essential for health promotion [39]. Nagarajan Maharajan et al. similarly characterized the therapeutic potential of flavonoid compounds in fatty liver disease associated with metabolic dysfunction [40]. Our study elaborated its role in IR from multiple perspectives, findings consistent with Yurtseven K et al. [41].

Anthocyanins constitute a specialized subclass of flavonoids, while flavanones – exemplified by hesperetin and naringenin – represent predominant bioactive constituents characteristic of citrus fruits. Both categories demonstrated formidable antioxidant efficacy through reactive oxygen species (ROS) and augmentation of endogenous antioxidant defenses, including superoxide dismutase and glutathione peroxidase. Investigations have established that anthocyanin-enriched dietary sources, such as blueberries or cherries, effectively suppress liver fat production, increase liver lipid oxidation and clearance, while modulating peroxisome proliferator-activated receptor expression [42]. Notably, the principal metabolite of anthocyanins, cyanidin-3-O-glucoside (C3G), has demonstrated Nrf2/ARE pathway activation, consequently amplifying cellular antioxidant response [43]. Chen, Qiu et al. 's study found that 12 weeks hesperidin administration substantially attenuated fasting blood glucose and fasting insulin levels in ob/ob mice, and enhanced insulin sensitivity. Additionally, this intervention diminished macrophage activation biomarkers Nos2 and Ptgs2 expression in mouse groin white adipose tissue (WAT) [44]. Chronic low-grade inflammation, driven by adipose tissue macrophage infiltration and pro-inflammatory cytokines such as tumor necrosis factor (TNF)-α and interleukin (IL)-6, exacerbates insulin resistance. Anthocyanins inhibit NF-κB activation by inhibiting i-κB kinase (IKK) phosphorylation, thereby reducing cytokine secretion [45]. These effects potentially operate through macrophage polarization modulation toward the anti-inflammatory M2 phenotype.

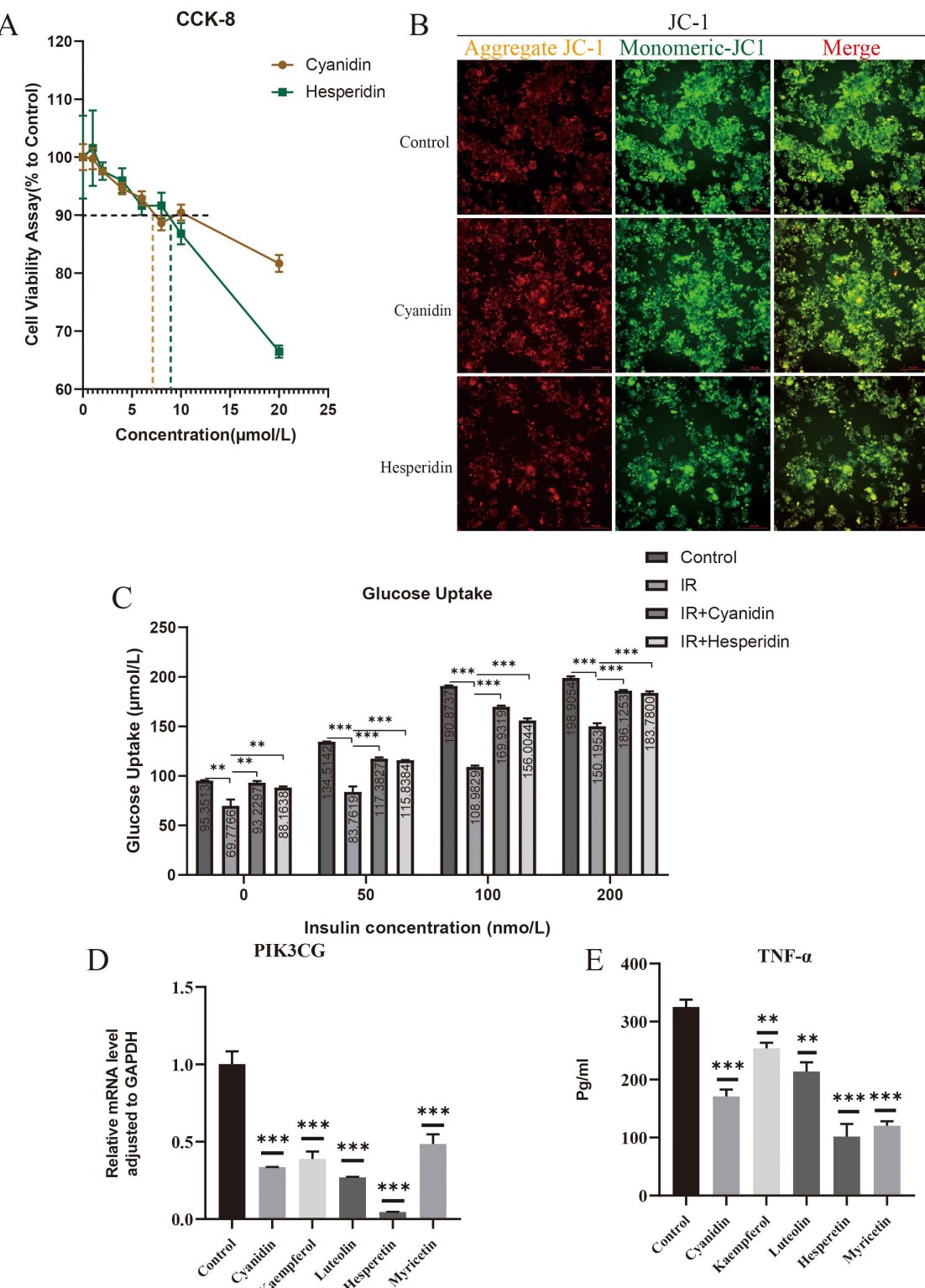

**Fig 7. Anthocyanidins and hesperidin improved insulin resistance of 3T3-L1 mature adipocytes. (A)** Effects of anthocyanidins and hesperidin on cell viability of 3T3-L1 mature adipocytes. **(B)** Effects of anthocyanidins and hesperidin on mitochondrial membrane potential of 3T3-L1 mature adipocytes. **(C)** Glucose uptake of 3T3-L1 mature adipocytes in control group, IR group and anthocyanidins and hesperidin stimulated groups under different insulin concentrations.

Accumulating evidence highlights anthocyanins' pivotal role in reshaping gut microbiota composition. In obese individuals, anthocyanin supplementation increases the abundance of Ackermansia mucophilus and bifidobacteria, strains correlated with improved intestinal barrier function and reduced endotoxemia [46].

Interestingly, our RCS results demonstrated that distinctive J-shaped associations between Isoflavones, Flavones, Flavan_3_ols with METS-IR. This relationship indicates that IR resistance risk exhibits progressive attenuation corresponding to increased consumption of these bioactive compounds within defined parameters; however, this beneficial trajectory plateaus upon exceeding optimal intake thresholds. We postulate that this phenomenon potentially reflects dose-dependent receptor saturation effect, wherein flavonoid substances improve insulin sensitivity by activating specific molecular targets [e.g., AMPK, peroxisome proliferator-activated receptor gamma (PPARγ)] or inhibiting inflammatory signaling (e.g., NF-κB). Nevertheless, these therapeutic mechanisms exhibit inherent threshold limitations. Hardie's investigation showed that AMPK phosphorylation attained maximum activation at specific concentrations, beyond which further dose escalation failed to elicit additional enzymatic stimulation [47]. Secondly, bioavailability and metabolic restriction constitute plausible explanations for these observations. Flavonoid oral bioavailability remains characteristically suboptimal, being contingent upon intestinal flora into active forms, while intestinal transporters [such as sodium-glucose transporter 1 (SGLT1)] possess limited uptake capacity for flavonoids. At high doses, the unabsorbed protocompound enters the colon directly and may be over-degraded by the flora into inactive metabolites [48–49]. Concurrently, our investigations revealed that Anthocyanidins, Flavanones, Flavonols and total flavones showed U-shaped curves with METS-IR. This may emanate from bidirectional regulation of anti-oxidation and pro-oxidation. Moderate flavonoid concentrations attenuate oxidative stress by clearing ROS, but excessive doses may trigger reverse effects: extremely high concentrations of flavonoids can generate reactive oxygen species in the presence of metal ions, counteracting their antioxidant effects [50]. Prolonged exposure to elevated concentrations may additionally suppress endogenous antioxidant enzyme expression, compromising the organism's intrinsic oxidative defense capabilities [51]. However, further research is needed in the future to explain the underlying mechanism.

To further elucidate the underlying mechanisms by which flavonoids ameliorate IR, we performed a comprehensive network pharmacological analysis. GO enrichment analysis showed that the neutrophil activation pathway, which regulates immunity, macrophage activation and fatty acid metabolism, constitutes the principal molecular mechanism of flavonoid anti-IR effects. KEGG analysis revealed that NF-κB, insulin pathway and AMPK pathway represent pivotal therapeutic targets for flavonoid substances. Accumulating evidence indicates that chronic low-grade inflammation of adipose tissue constitutes a hallmark feature of diverse metabolic diseases, including IR. During inflammation, lipid metabolism in adipose tissue becomes progressively dysfunctional, characterized by extensive pro-inflammatory cell infiltration and high cytokine concentration, and specific inflammatory signals such as the NF-κB pathway are an important part of this process [52]. Functioning as an energy sensor, AMPK induces AMPK and other molecules such as PI3K/Akt, NADPH oxidase 4 (NOX4) and NF-κB in the course of IR, enhancing cellular glucose uptake capacity [53]. Our study found that genes exhibiting diminished expression were predominantly enriched within AMPK pathway, suggesting that adipose tissue in IR patients has lost the protective effect of AMPK. Overall, the enrichment analysis of these differentially expressed genes demonstrated the involvement of immune cell-mediated inflammation and abnormal fatty acid metabolism in adipose tissue of IR patients, as well as the failure of protective factors such as AMPK. Through integrated analysis of all insulin resistance-related differentially expressed genes and their intersection with 12 dietary flavonoid-associated genes, we identified 5 overlapping key genes. Among them, PIK3 CG exhibited the most ubiquitous distribution pattern, followed by MMP9 and SYK, indicating that PIK3 CG, MMP9 and SYK likely represent universal targets for various dietary flavonoid improvements in IR process, which aligns with previous studies [54–55]. The PPI network analysis demonstrated robust interactions among PIK3 CG, MMP9, SYK and PRKCB. KEGG and GO enrichment analysis revealed that these four genes were significantly enriched in inflammatory response and NF-κB signaling pathway, corroborating the pathological foundation of IR caused by low-grade inflammation in adipose tissue. Previous investigations demonstrated that the abundance

of infiltrating pro-inflammatory macrophages was significantly reduced in adipose tissue and liver of obese mice with Pik3 cg (-/-), leading to suppression of the inflammatory response in these tissues, establishing PIK3 CG as a promising therapeutic target for insulin resistance and type 2 diabetes [56]. According to the study of Li S et al., inhibition of PIK3 CG/p65/MMP9 signaling can alleviate LPS-induced inflammatory damage [57]. IR exhibits close correlation with chronic low-grade inflammation resulting from abnormal macrophage infiltration into adipose tissue. Therefore, the Syk-PI3K axis-mediated polarization of immunosuppressive macrophages to ameliorate pro-inflammatory phenotype macrophage infiltration may represent a novel therapeutic strategy for IR improvement [58]. Impaired mitochondrial respiratory activity can precipitate IR development, frequently manifesting as a declined mitochondrial membrane potential. Studies have shown that the activation of PRKCB negatively regulates mitochondrial energy status and inhibits autophagy. PRKCB inhibition can increase mitochondrial membrane potential [59–60], potentially improving cellular energy metabolism and subsequently alleviating IR. Our study used molecular docking techniques to establish strong correlations between PIK3 CG, MMP9, SYK, PRKCB and 12 dietary flavonoids, suggesting that these genes may serve as key targets for dietary flavonoid improvement of IR.

Based upon NHANES study findings, anthocyanidins and hesperidin were selected as representative dietary flavonoid compounds for in vitro experimental validation. Serving as a critical biomarker for mitochondrial function and cellular energetic status, mitochondrial membrane potential represents an indispensable parameter in metabolic assessment. Previous studies have established that IR of adipocytes is manifested as impairment of mitochondrial membrane potential [61]. For the purpose of examining dietary flavonoid-mediated restoration of mitochondrial membrane potential, JC-1 staining was implemented. Experimental data revealed that the mitochondrial membrane potential of 3T3-L1 adipocytes in the IR group exhibited elevated mitochondrial membrane potential subsequent to anthocyanidin or hesperidin treatment, implying these bioactive compounds possess IR-mitigating properties. According to Cremonini et al., anthocyanins counteracted high-fat diet-induced modifications in mitochondrial dynamics, biogenesis, and thermogenesis in subcutaneous white adipose tissue (sWAT) of mice, representing a plausible mechanism for their improvement in IR [62]. Through Glucose Uptake Fluorometric Assay, anthocyanidins or hesperidin administration yielded partial glucose uptake recovery compared to the IR group at identical insulin concentrations, though values remained suboptimal relative to control group. This phenomenon likely correlates with dietary flavonoid duration and dose. The aforementioned experimental evidence conclusively establishes that anthocyanidins or hesperidin successfully reverse insulin resistance and mitochondrial impairment in 3T3-L1 adipocytes.

A critical consideration in interpreting our findings concerns whether the concentrations employed in cell experiments (7 µmol/L anthocyanidins and 9 µmol/L hesperidin) are physiologically relevant. While typical plasma flavonoid levels following dietary intake range between 0.1–1 µmol/L, several lines of evidence substantiate the biological significance of our chosen doses. First, although flavonoids exhibit relatively low systemic bioavailability, they may accumulate preferentially in metabolically active tissues like adipose through enterohepatic recycling and lipophilic partitioning. This tissue-specific accumulation could yield localized concentrations exceeding plasma levels by 5–10 fold. Second, clinical trials using pharmacologic doses of purified anthocyanin extracts (500–1000 mg/day) report peak plasma concentrations of 5–10 µmol/L, directly overlapping with our experimental range. Third, even when parent compounds undergo rapid metabolic transformation, their bioactive derivatives demonstrate insulin-sensitizing effects at nanomolar concentrations. From a translational perspective, while achieving sustained 7–9 µmol/L systemic concentrations through typical dietary patterns remains challenging, targeted nutritional interventions could approach these levels. For instance, consumption of 300 g of blueberries, which provides 500 mg anthocyanins or nutraceutical preparations utilizing citrus peel (rich in hesperidin) has demonstrated the capacity to transiently elevate plasma concentrations into the pharmacologically active range. These observations suggest our experimental conditions may model high-dose supplementation scenarios or localized tissue accumulation.

Our study presents several limitations: First, this cross-sectional design precluded determination of causal relationships between flavonoid intake and IR. Second, dietary flavonoid intake quantification based on 24-hour dietary recall,

potentially introduces recall bias. Thirdly, analysis of total flavonoids-IR associations was constrained by the completeness of exposure measurements in the NHANES database, preventing comprehensive resolution of complex correlations and potential interactions among various subcategories. Future research should integrate multi-center cohort data and apply dedicated methods for exposure mixtures to elucidate synergistic/antagonistic mechanisms of flavonoid subclasses. Concurrently, this study relied predominantly on online data sets for analysis and only conducted experimental validation exclusively in vitro. Future studies should employ advanced techniques to study PIK3 CG, MMP9, SYK and PRKCB expression levels in populations and examine dietary flavonoid effects on these target genes in humans. Despite these limitations, our integrated approach – combining population data, network pharmacology, and functional validation – provides compelling evidence that dietary flavonoids, particularly anthocyanidins and flavanones, may serve as effective adjunct therapies for IR. The identification of PIK3 CG, MMP9 and SYK as conserved molecular targets offers mechanistic insights for developing targeted nutritional interventions.

## 5. Conclusion

Our results demonstrate that flavonoid intake exhibits an inverse associateion IR risk. This highlights the potential significance of flavonoid intake in obesity and IR management, providing valuable information for tailoring nutritional intervention strategies for IR treatment. However, to establish definitive causal relationships between flavonoids and IR, future large-scale prospective cohort studies or randomized controlled trials are warranted.

## Supporting information

**S1 Fig. Molecular docking of PIK3 CG with dietary flavonoids.**
(TIFF)

**S2 Fig. Molecular docking of MMP9 and SYK with dietary flavonoids.**
(TIFF)

**S3 Fig. Molecular docking of Cyanidin and Hesperetin with dietary flavonoids.**
(TIFF)

**S4 Fig. The results of KEGG and GO enrichment analysis of hub genes.**
(TIFF)

**S1 Table. Subgroup analysis between anthocyanidins and METS-IR.** Very low: [0,0.15]mg/d, Low: (0.15,2.36]mg/d, Moderate: (2.36,13.6]mg/d, High: > 13.6mg/d Adjusted by age, sex, race, education, PIR, smoking status, drinking status, caloric intake, physical activity, coronary heart disease, and hypertension.
(XLSX)

**S2 Table. Subgroup analysis between flavanones and METS-IR.** Very low: [0,0.1]mg/d, Low: (0.1,1]mg/d, Moderate: (1,20]mg/d, High: > 20mg/d.Adjusted by age, sex, race, education, PIR, smoking status, drinking status, caloric intake, physical activity, coronary heart disease, and hypertension.
(CSV)

**S3 Table. Test for the interaction between flavonoid subclasses and total flavonoid intake.** Adjusted by age, sex, race, education, PIR, smoking status, drinking status, caloric intake, physical activity, coronary heart disease, and hypertension. Total Sum of all 29 flavonoids: [0,50]mg/d, (50,200]mg/d, (200,500]mg/d, > 500mg/d. Total Anthocyanidins: [0,0.15]mg/d, (0.15,2.36]mg/d, (2.36,13.6]mg/d, > 13.6mg/d. Total Isoflavones: [0,0]mg/d, (0.001,0.015]mg/d, (0.02,0.1] mg/d, > 0.1mg/d. Total Flavan_3_ols: [0,5.3]mg/d, (5.03,17]mg/d, (17,170] mg/d, > 170mg/d. Total Flavanones: [0,0.1]mg/d,

(0.1,1]mg/d, (1,20]mg/d, > 20mg/d. Total Flavones: [0,0.2]mg/d, (0.2,0.6]mg/d, (0.6,1.2]mg/d, > 1.2mg/d. Total Flavonols: [0,8]mg/d, (8,14]mg/d, (14,24]mg/d, > 24mg/d.
(DOCX)

**S4 Table. List of 29 types of flavonoid compounds.**
(DOCX)

**S1 Data. STROBE-nut_checklist.docx Reporting checklist for observational studies in nutritional epidemiology.**
(DOCX)

## Author contributions

**Conceptualization:** Yanhong Liu, Junde Zhao.

**Data curation:** Xiaohui Sui, Yanhong Liu, Junde Zhao, Zuocheng Wang.

**Investigation:** Xiaohui Sui.

**Software:** Junde Zhao, Zuocheng Wang.

**Supervision:** Zuocheng Wang.

**Validation:** Xiaohui Sui.

**Visualization:** Xiaohui Sui.

**Writing – original draft:** Xiaohui Sui, Yanhong Liu, Junde Zhao, Guiju Zhang.

**Writing – review & editing:** Guiju Zhang.

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
