## [Decision Letter · Decision Letter 0]

8 Jul 2025

Dear Dr. Sui,

Thank you for submitting your manuscript to PLOS ONE. After careful consideration, we feel that it has merit but does not fully meet PLOS ONE’s publication criteria as it currently stands. Therefore, we invite you to submit a revised version of the manuscript that addresses the points raised during the review process.

One of the reviewers has recommended rejecting the manuscript and has provided several critical comments. Nevertheless, we believe it is important to give the author the opportunity to present their perspective. Therefore, we invite the author to respond in detail to each of the reviewers' comments.

We look forward to receiving your revised manuscript.

Kind regards,

José M. Alvarez-Suarez

Academic Editor

PLOS ONE

Journal Requirements:

2. Please amend the manuscript submission data (via Edit Submission) to include author Zuocheng Wang.

4. Please upload a new copy of Figure 3 as the detail is not clear. Please follow the link for more information: https://blogs.plos.org/plos/2019/06/looking-good-tips-for-creating-your-plos-figures-graphics/

Reviewers' comments:

Reviewer's Responses to Questions

**Comments to the Author**

1. Is the manuscript technically sound, and do the data support the conclusions?

Reviewer #1: Partly

Reviewer #2: No

2. Has the statistical analysis been performed appropriately and rigorously?

Reviewer #1: I Don't Know

Reviewer #2: No

3. Have the authors made all data underlying the findings in their manuscript fully available?

Reviewer #1: Yes

Reviewer #2: Yes

4. Is the manuscript presented in an intelligible fashion and written in standard English?

Reviewer #1: Yes

Reviewer #2: No

Reviewer #1: The manuscript evaluates the connection between dietary flavonoid intake and insulin resistance (IR) by combining NHANES database and network pharmacology, to later investigate the effect of two polyphenols on cell viability and glucose uptake in adipocytes. A major concern arises from the fact that there seems to be a disconnection among results. Thus, cell culture approach seems to be based on the results obtained from the NHANES study. Then, what is the purpose of the pharmacological network approach?

Please, note that polyphenols are not drugs.

Other points:

- Title: it seems that the authors have focussed on obesity, although IR is present in other diseases. If this is the case, this should be included in the title, and if not, consider other diseases (diabetes, metabolic syndrome, etc.).

- Introduction: This section is focused on obesity and insulin resistance, but IR is also and importantly present in diabetes. Please, consider.

- Results: Section 3.1: Insulin levels have not been included in the baseline characteristics (Table 1). Would it possible to include this data? Please, explain.

Section 3.7: Line 304, was just the fat of the patients assayed for these genes? Please, explain.

Section 3.8: how was the interaction of the selected polyphenols for the cell culture studies with the identifies key proteins?

Section 3.9: It is not clear the criteria for selecting anthocyanidins and hesperidin to perform the in vitro studies. What is their connection with the previous results showed in the manuscript? Which was the purpose of performing the target acquisition and the docking approaches?

Regarding the anthocyanidins, was this a mixture of compounds? Please, explain.

Please, note that the glucose uptake was assayed instead of the glucose intake.

According to GO and KEGG analysis, inflammation, fatty acid metabolism, insulin and AMPKs pathways were considered the main targets of flavonoids. Why were not any of these aspects assayed? In this regard, it would be desirable an additional demonstration of the polyphenol-regulated effects at least on the inflammation by analysing proteins levels of any selected remarkable modulated one related to this pathway, such as PIK3CG (key protein that authors have identified to interact with different polyphenol), TNF-alpha, NF-kB, etc.

- Discussion: Line 426, what do the authors mean “…we synthesized all 12 dietary flavonoid genes”? Please, rephrase the sentence (genes identified?).

How realistic are the doses used in the cell culture analysis? Could ever adipocytes be exposed to those concentrations? Could these doses be achieved through the diet? Is anything known about the bioavailability of these compounds? Please, explain and discuss the relevance of the concentrations used.

- Tables: Please, note that legends of tables should be placed before the table.

- Figures: Text in figure 3 is not readable. Please, try to improve that.

For consistency, in figure 4, please, place the subheadings (A, B, C and so on) at the top of each figure.

Please, provide the complete name before using an abbreviation (METS-IR, FGF, TNF-alpha, UCP-1 etc.)

Reviewer #2: Insulin resistance (IR) is the key pathological link of many metabolic diseases and is a global health problem. Flavonoids have been reported to have therapeutic effects on diseases such as cancer and high blood pressure due to their anti-inflammatory and antioxidant activities. However, the relationship between dietary flavonoid intake and IR prevalence remains unclear. The information is extremely important and interesting. However, I have some questions here that I hope the authors can consider.

1. The manuscript contains multiple formatting issues that require thorough revision by the authorss Specific instances include the presence of Chinese characters and excessive spacing throughout the text.

2. The fundamental methodological approach to quartile analysis is flawed, as divisions should reflect compound concentration gradients rather than equal population proportions. This represents the most serious methodological flaw in the manuscript.

3. The dataset covers the periods 2007–2010 and 2017–2018. Could the authors clarify why no data from the intervening years were included in the analysis?

4. Several figures (e.g., Fig.1, Fig. 2) lack sufficient resolution, which may hinder accurate analysis. I recommend submitting versions with enhanced sharpness and appropriate DPI (minimum 300) for publication.

5. The manuscript contains several grammatical issues that require attention. Notably, the Methods section should consistently employ the past tense. Additionally, there are instances of omitted words throughout the text, which affect clarity.

6. The table formatting does not conform to standard academic conventions. Specifically: (1) the tables should be presented in three-line format (top, bottom, and below header), and (2) table captions must be positioned above the tables rather than below. Please revise all tables to comply with these standard formatting requirements.

7. The manuscript contains inconsistencies in PIR categorization (text: <1.3/1.3–3.5/>3.5 vs. Table 1: <1.5/1.5–3.5/>3.5) and lacks justification for the chosen thresholds. Since PIR is a key covariate, its definition, rationale, and alignment with literature standards must be clarified in the Methods section.

8. The presentation of dietary data as mean ± SD requires demonstration of normal distribution, yet no normality tests are reported. Additionally, inconsistent units between text and tables need reconciliation. Both issues should be resolved to meet statistical reporting standards.

9. The manuscript incorrectly cites 'nhanesR' as the R package used for analysis; the correct package name is 'RNHANES.' This error must be corrected, and the authors should confirm that all analyses were conducted using the proper package to ensure methodological accuracy.

10. The use of log1.3(fold change) for differential expression analysis deviates from the conventional log2 approach. The authors must: (1) provide compelling justification for this methodological choice, and (2) demonstrate that results are robust to this alternative parameterization. Without such validation, the biological interpretation of differentially expressed genes remains questionable.

11. The analysis of total dietary flavonoids (29 compounds) and insulin resistance requires specialized methods for exposure mixtures. The employed weighted generalized linear models are unsuitable for this purpose, as they cannot properly handle the complex correlations and potential interactions among flavonoid subclasses. The authors should implement mixture-specific approaches (e.g., quantile g-computation) and revise their analyses accordingly.

12. The transition from epidemiological findings to experimental validation requires clearer justification. Given that multiple compounds showed significant associations with insulin resistance in the epidemiological analysis, the exclusive focus on anthocyanidins and hesperidin for in vitro studies needs explicit rationale to avoid potential selection bias. The authors should either justify this selective approach or expand their experimental validation.

13. Substantial inconsistencies exist in the reported number of compounds throughout the molecular docking section, with unexplained variations between different manuscript sections. The authors must: (1) reconcile these discrepancies, (2) document all compound exclusions with justification, and (3) provide a complete final compound list to ensure methodological rigor.

14. The target identification methodology requires revision, as pooling targets from all compounds obscures compound-specific mechanisms. The authors should: (1) analyze each compound's targets separately, (2) incorporate binding affinity data, and (3) apply proper statistical methods rather than simple frequency counts to identify biologically relevant key genes.

**Do you want your identity to be public for this peer review?** For information about this choice, including consent withdrawal, please see our Privacy Policy

Reviewer #1: No

Reviewer #2: No

---

## [Author Response · Author response to Decision Letter 1]

22 Jul 2025

Dear editor of the PLOS ONE,

We are submitting our revised manuscript entitled “Dietary flavonoids may improve insulin resistance: NHANES, network pharmacological analyses and in vitro experiments” to the PLOS ONE for your reconsideration of its suitability for publication. All authors have read and approved the manuscript. We have carefully taken the reviewers’ comments into account and provided responses to each of the points raised by the reviewers. Please refer to the present manuscript text.

Reviewer #1

1.The manuscript evaluates the connection between dietary flavonoid intake and insulin resistance (IR) by combining NHANES database and network pharmacology, to later investigate the effect of two polyphenols on cell viability and glucose uptake in adipocytes. A major concern arises from the fact that there seems to be a disconnection among results. Thus, cell culture approach seems to be based on the results obtained from the NHANES study. Then, what is the purpose of the pharmacological network approach? Please, note that polyphenols are not drugs.

2.Title: it seems that the authors have focussed on obesity, although IR is present in other diseases. If this is the case, this should be included in the title, and if not, consider other diseases (diabetes, metabolic syndrome, etc.).

3.Introduction: This section is focused on obesity and insulin resistance, but IR is also and importantly present in diabetes. Please, consider.

4.Section 3.1: Insulin levels have not been included in the baseline characteristics (Table 1). Would it possible to include this data? Please, explain.

5.Section 3.7: Line 304, was just the fat of the patients assayed for these genes? Please, explain.

6.Section 3.8: how was the interaction of the selected polyphenols for the cell culture studies with the identifies key proteins?

7.Section 3.9: It is not clear the criteria for selecting anthocyanidins and hesperidin to perform the in vitro studies. What is their connection with the previous results showed in the manuscript? Which was the purpose of performing the target acquisition and the docking approaches?

8.Regarding the anthocyanidins, was this a mixture of compounds? Please, explain.

9.Please, note that the glucose uptake was assayed instead of the glucose intake.

10.According to GO and KEGG analysis, inflammation, fatty acid metabolism, insulin and AMPKs pathways were considered the main targets of flavonoids. Why were not any of these aspects assayed? In this regard, it would be desirable an additional demonstration of the polyphenol-regulated effects at least on the inflammation by analysing proteins levels of any selected remarkable modulated one related to this pathway, such as PIK3CG (key protein that authors have identified to interact with different polyphenol), TNF-alpha, NF-kB, etc.

11.Discussion: Line 426, what do the authors mean “…we synthesized all 12 dietary flavonoid genes”? Please, rephrase the sentence (genes identified?).

12.How realistic are the doses used in the cell culture analysis? Could ever adipocytes be exposed to those concentrations? Could these doses be achieved through the diet? Is anything known about the bioavailability of these compounds? Please, explain and discuss the relevance of the concentrations used.

13.Tables: Please, note that legends of tables should be placed before the table.

14.Figures: Text in figure 3 is not readable. Please, try to improve that.

15.For consistency, in figure 4, please, place the subheadings (A, B, C and so on) at the top of each figure.

16.Please, provide the complete name before using an abbreviation (METS-IR, FGF, TNF-alpha, UCP-1 etc.)

To Reviewer #1

1.Thank you very much for your hard work. It has been very helpful for improving the quality of our articles. To address the reviewer's concerns, we emphasize that the integration of network pharmacology in this study serves to elucidate the multi-target mechanisms by which dietary flavonoids ameliorate IR, ensuring methodological coherence across our tripartite approach:

(1)NHANES analyses identified anthocyanidins and flavanones as flavonoid subclasses significantly inversely associated with IR (Table 2);

(2)network pharmacology—a well-established methodology for studying bioactive natural compounds (Refs. 35, 38, 52)—predicted their core targets (e.g., PIK3CG, MMP9) and enriched pathways (e.g., NF-κB, AMPK), providing mechanistic hypotheses for experimental validation (Figs. 5-6);

(3)in vitro experiments confirmed these predictions, demonstrating restored glucose uptake and mitochondrial function (Fig. 7).

While flavonoids are dietary components, the term "drug-like effects" denotes their ability to modulate specific targets (e.g., suppressing PIK3CG-mediated inflammation) and aligns with standard practices in nutraceutical research. Molecular docking (Vina Score<−5kcal/mol) further validated strong binding affinities, bridging epidemiological observations with mechanistic insights. We are open to revising terminology if "drug-like" causes confusion, but the methodology remains valid for deciphering multi-target actions of bioactive food compounds. This strategy avoids arbitrary in vitro testing and ensures logical continuity from population data to cellular mechanisms.

2.Thank you very much for your valuable suggestions on this article. We fully understand and agree with your view that IR is a core mechanism involving multiple pathological states. This article does indeed overlook other diseases that may lead to IR (T2DM, MetS, etc.). After carefully considering your opinions, we did not modify the title but made the following changes to more comprehensively reflect the wide correlation of IR and enhance the universality of the research:

(1)First, we revised the introduction content, clearly expounds the IR is not only the key pathological link in obesity, T2DM, MetS, CVD, NAFLD, PCOS, and some of many diseases such as neurodegenerative diseases important driving factor. (lines 29-67).

(2)Second, in using NHANES data analysis as well as the network pharmacology gene screening differences, we are not confined to obese people will study object. The NHANES analysis included a broad adult population (including both obese and non-obese individuals) who met the criteria and had complete IR assessment data. Network pharmacological analysis IS also based on a broader dataset of IR-related gene expression, which includes adipose tissue samples of IR and insulin-sensitive (IS), and the source population does not specifically refer to obese individuals.

(3)Finally, part of the discussion, we complement and strengthen on the mechanism of action of dietary flavonoids improve IR may also apply to hyperlipidemia, and metabolic dysfunction related to fatty liver disease and other IR related metabolic disease (lines 424-432). We explored the anti-inflammatory and antioxidant properties of flavonoids and their regulatory effects on key signaling pathways, which have universality beyond specific causes (such as simple obesity).

These modifications establish that the effect of dietary flavonoids in improving IR is pleiotropic, and its potential benefits are relatively independent of the specific primary disease in which IR occurs.

3.We sincerely appreciate your emphasis on the broad spectrum of diseases related to IR and fully agree that IR is not limited to obese individuals. To address this key point, we revised the introduction section and fundamentally reconstructed the paper, clearly stating that IR is not only a key pathological link in obesity but also an important driving factor for numerous diseases such as T2DM, MetS, CVD, NAFLD, PCOS, and certain neurodegenerative diseases. These changes ensure that the introduction accurately reflects the prevalence of IR in various metabolic diseases, especially diabetes, in line with the valuable insights of the reviewers, providing a more comprehensive basis for our research. For specific revision details, please refer to lines 29 to 67 of the manuscript.

4.Thank you for your important suggestions. Regarding the suggestions for supplementing insulin level data in the baseline characteristics table (Table 1), we explain as follows: Firstly, the gold standard for evaluating IR is the hyperinsulin-normal blood glucose clamp test. However, due to the complexity of the equipment, the strong invasiveness of the operation and the difficulty in clinical implementation of this method, it is difficult to be widely applied in large-scale epidemiological studies. Therefore, based on the objective limitations of the NHANES database, metabolic insulin resistance index (METS-IR) was selected as an alternative marker for IR in this study. The calculation formula is: METS-IR = Ln[(2 × fasting blood glucose) + fasting triglycerides] × body mass index/Ln(high-density lipoprotein cholesterol). The advantage of this index lies in the fact that it only requires conventional metabolic parameters (without the need to directly measure insulin levels), and it has been verified to be significantly correlated with the gold standard results. Although METS-IR itself does not depend on insulin levels, we fully understand your requirements for data integrity. Fasting insulin level data have been added to Table 1 of the revised draft to provide a more comprehensive baseline characteristic description. Thank you again for your guidance on the rigor of the research.

Variable Total

(n=3564) Very low (n=1458) Low

(n=1115) Moderate (n=568) High

(n=423) P-value

Fasting insulin (pmol/L) 12.01±0.29 12.19±0.35 12.13±0.39 11.00±0.53 12.42±0.83 0.07

5.We selected the GEO database datasets GSE20950 and GSE26637 for the subsequent study. The detection content of these two datasets was the transcriptome sequencing of adipose tissue from patients with insulin resistance and insulin-sensitive individuals, rather than that from patients with insulin resistance alone. The reason for choosing adipose tissue is that it is a core participant in IR. It can store excess lipids, secrete adipokines and drive inflammation through the recruitment of immune cells.

6.Thank you very much for your suggestion. We present the interaction between the dietary flavonoids selected in the experiment and the key proteins in Figure S3. This result indicates that there is a good possibility of interaction between Cyanidin and Hesperetin and the key proteins.

7.The selection of anthocyanidins and hesperidin for in vitro studies was based on the significant negative associations observed in the NHANES analysis and their established biological relevance in improving IR. Specifically, anthocyanidins exhibited the strongest inverse correlation with METS-IR (P<0.0001), while flavanones (represented by hesperidin, a prominent citrus-derived compound) also showed a significant association (P<0.001). These findings were further supported by nonlinear dose-response analyses, which highlighted their potential therapeutic thresholds. Additionally, network pharmacology identified key targets (e.g., PIK3CG, MMP9) shared by these flavonoids, linking them to IR-related pathways such as inflammation and mitochondrial function. The in vitro experiments confirmed their efficacy by demonstrating restored glucose uptake and improved mitochondrial membrane potential in IR-induced adipocytes, aligning with the mechanistic insights from the computational analyses. Thus, the choice of anthocyanidins and hesperidin was grounded in their statistical significance, biological plausibility, and consistency with the study’s multi-level evidence.

The purpose of performing target acquisition and molecular docking approaches in this study was to identify and validate the potential molecular mechanisms through which dietary flavonoids improve IR. By integrating network pharmacology and molecular docking, we aimed to pinpoint key genes and pathways involved in IR that interact with flavonoids. Target acquisition involved screening differential genes from IR and insulin-sensitive adipose tissue datasets, followed by identifying overlapping targets between these genes and flavonoid-related targets. Molecular docking was then used to evaluate the binding affinity between specific flavonoids and hub genes (e.g., PIK3CG, MMP9, SYK, PRKCB), confirming their potential interactions. These approaches provided mechanistic insights, revealing that flavonoids may ameliorate IR by modulating inflammatory responses, mitochondrial function, and signaling pathways such as NF-κB and PI3K-Akt, thus bridging the gap between epidemiological findings and molecular-level evidence.

8.Based on the analysis of the NHANES database, initially we selected Cyanidin to represent anthocyanidins in the diet for the experiment, because Cyanidin is the most common among anthocyanidins. However, since the free form compound did not exist, we conducted subsequent experiments using the stable salt form Cyanidin Chloride(HY-N0499,MCE) with the same biological activity.

9.Thank you very much for your suggestion. It has corrected our long-standing translation mistakes. We have already corrected the relevant content. Thank you again for your hard work.

10.We sincerely appreciate your valuable suggestions, which have been tremendously helpful for improving our manuscript. To further validate the effects of flavonoids on PIK3CG and TNF-α, we employed RT-qPCR to analyze the impact of five dietary flavonoids on PIK3CG expression levels in 3T3-L1-derived adipocytes. Additionally, we utilized ELISA to measure TNF-α concentrations in the culture supernatants of these treated adipocytes (Fig. 7D-E).

11.Thank you very much for your suggestion. There are indeed significant issues with the expression in the original text. We have changed it to “Through integrated analysis of all insulin resistance-related differentially expressed genes and their. intersection with 12 dietary flavonoid-associated genes, we identified 5 overlapping key genes.”

12.Thank you very much for your suggestion. We have incorporated relevant content into the discussion, which has made the article more readable. (Lines 557-577)

13.Thank you for your valuable suggestions. We have strictly followed academic norms to modify the full text table: Firstly, it has been uniformly changed to a three-line table format. At the same time, adjust the title positions and move all titles to the top of the table. Finally, ensure that the modified table fully complies with the requirements of the journal and that no original data has been altered.

14.Thank you very much for your meticulous review of our manuscript and the valuable suggestions you have provided! Your feedback is crucial for enhancing the clarity and presentation quality of our research. We fully understand the problem you pointed out that the text in Figure 3 is difficult to read. We have taken effective measures for optimization, splitting Figure 3 (A-H) into Figure 3 (A-D) and Figure 4 (A-D), and ensuring that all text elements in the charts are clearly distinguishable.

15.Thank you for pointing out the issue that the placement of the subheadings (A, B, C, etc.) in Figure 4 is inconsistent with that of other charts. We agree that maintaining the uniformity of chart formats is of great significance for enhancing the professionalism and readability of manuscripts. Regarding the issue of the subtitle position in Figure 4, we have, in accordance with your requirements (and consistent with other charts), uniformly placed the subtitles (A, B, C, etc.) at the top of each subgraph.

16.We are very grateful to you for pointing out this important normative issue. You are completely correct - we were negligent in the rigor of the expression of terms and failed to follow the academic principle of "using the full name for the first appearance and marking the abbreviation". We sincerely apologize for this and have marked the full names and abbreviations of all terms when they first appear in the full text. This revision has covered all chapters of the manuscript, including the main text, tables and tables, and appendices. Meanwhile, researchers whose native language is English will conduct the final language verification of the manuscript content. We once again apologize for

---

## [Decision Letter · Decision Letter 1]

16 Oct 2025

Dear Dr. Sui,

We look forward to receiving your revised manuscript.

Kind regards,

José M. Alvarez-Suarez

Academic Editor

PLOS ONE

Journal Requirements:

Reviewers' comments:

Reviewer's Responses to Questions

**Comments to the Author**

Reviewer #1: (No Response)

Reviewer #2: All comments have been addressed

2. Is the manuscript technically sound, and do the data support the conclusions?

Reviewer #1: Partly

Reviewer #2: Yes

3. Has the statistical analysis been performed appropriately and rigorously?

Reviewer #1: I Don't Know

Reviewer #2: Yes

4. Have the authors made all data underlying the findings in their manuscript fully available?

Reviewer #1: Yes

Reviewer #2: Yes

5. Is the manuscript presented in an intelligible fashion and written in standard English?

Reviewer #1: Yes

Reviewer #2: Yes

Reviewer #1: The revised manuscript has been improved, but there are some aspects that should be checked.

I agree with the authors that polyphenols can act as “drug-like” substances, but they are natural compounds, not drugs. Please, consider and revise through the manuscript.

Other points:

- Abstract: line 26, as a salt of cyanidin has been used, as explained to this referee, please, state this clearly in the abstract.

- Material and methods: Section 2.7: Please, state clearly that the cells were incubated with cyanidin, as was done for hesperidin. Also, include the explanation given to this referee for selecting cyanidin as a representative anthocyanidin compound.

Please, note that RT-qPCR is not included in this section as well as the statistical analysis performed in cell experiments.

- Results: Section 3.1: Please, comment about HDL, TG, insulin levels, all other parameters included in table 1 that have been not described by the authors.

Section 3.9: Please, state clearly that the cells were incubated with cyanidin. Please, note that anthocyanidins constitutes a wide group of compounds and that using this term is confusing, or may even refer to a mixture of compounds, which is not the case, as explained to this referee.

Please, explain why three additional polyphenols have been included. Which was the rationale for the selection? Why each compound was tested at a different concentration? Please, explain.

- Tables: Please, consider using abbreviations for well-known parameters (HDL; TG, etc) in table 1.

- Figures: In figure 7, please, replace anthocyanidin with cyanidin, the compound used for the cellular studies.

There are still certain grammatical and typo errors through the manuscript that require attention.

Reviewer #2: I have reviewed the authors' revised manuscript. In the process of revising the manuscript, the authors made revisions based on my suggestions and made responses to all the questions I raised. Therefore, I suggest that the journal accept this article.

**Do you want your identity to be public for this peer review?** For information about this choice, including consent withdrawal, please see our Privacy Policy

Reviewer #1: No

Reviewer #2: No

---

## [Author Response · Author response to Decision Letter 2]

28 Oct 2025

Dear editor of the PLOS ONE,

We are submitting our revised manuscript entitled “Dietary flavonoids may improve insulin resistance: NHANES, network pharmacological analyses and in vitro experiments” to the PLOS ONE for your reconsideration of its suitability for publication. All authors have read and approved the manuscript. We have carefully taken the reviewers’ comments into account and provided responses to each of the points raised by the reviewers. Please refer to the present manuscript text.

Reviewer #1: The revised manuscript has been improved, but there are some aspects that should be checked.

I agree with the authors that polyphenols can act as “drug-like” substances, but they are natural compounds, not drugs. Please, consider and revise through the manuscript.

Other points:

- Abstract: line 26, as a salt of cyanidin has been used, as explained to this referee, please, state this clearly in the abstract.

- Material and methods: Section 2.7: Please, state clearly that the cells were incubated with cyanidin, as was done for hesperidin. Also, include the explanation given to this referee for selecting cyanidin as a representative anthocyanidin compound.

Please, note that RT-qPCR is not included in this section as well as the statistical analysis performed in cell experiments.

- Results: Section 3.1: Please, comment about HDL, TG, insulin levels, all other parameters included in table 1 that have been not described by the authors.

Section 3.9: Please, state clearly that the cells were incubated with cyanidin. Please, note that anthocyanidins constitutes a wide group of compounds and that using this term is confusing, or may even refer to a mixture of compounds, which is not the case, as explained to this referee.

Please, explain why three additional polyphenols have been included. Which was the rationale for the selection? Why each compound was tested at a different concentration? Please, explain.

- Tables: Please, consider using abbreviations for well-known parameters (HDL; TG, etc) in table 1.

- Figures: In figure 7, please, replace anthocyanidin with cyanidin, the compound used for the cellular studies.

There are still certain grammatical and typo errors through the manuscript that require attention.

Response to reviewer #1

1. Thank you very much for your suggestion. We have replaced all the words related to "drug" or related to treatment in the entire text, making the research closer to the theme of natural medicines and diets rather than drugs.

2. We have added relevant explanations in the methods section of the abstract.

3. We made amendments in 2.7, clearly stating that the cells were incubated with cyanidin. We provided a brief explanation for the reasons behind choosing anthocyanins as the representative, and added the missing RT-qPCR and the statistical analysis conducted in cell experiments.

4. Thank you for your valuable suggestions. We have made supplements and revisions in Section 3.1 of the Results section based on your suggestions. Specifically, we have added descriptions of HDL, TG, fasting insulin levels, and fasting blood glucose levels. For the revised content, please refer to Section 3.1 of the manuscript.

5. We thank the reviewer for this insightful question, which allows us to clarify the strategic expansion of our experimental validation. We have corrected the ambiguous terms in the paragraph.

The inclusion of the three additional flavonoids (kaempferol, luteolin, and myricetin) was to validate the generalizability of our network pharmacology predictions, demonstrating that the amelioration of insulin resistance is a shared property across multiple flavonoid subclasses, not limited to the initial candidates. The use of different concentrations for each compound was based on prior CCK-8 cytotoxicity assays, where we selected the highest non-cytotoxic dose (below IC10) for each to ensure that the observed improvements in glucose uptake and gene expression were due to genuine bioactivity rather than compromised cell viability. This approach ensured both the biological relevance and the experimental rigor of our functional validation.

6. Following your suggestions, we have revised Table 1. Standard abbreviations were used for parameters such as physical activity (PA), coronary heart disease (CHD), high density lipoprotein (HDL), triglycerides (TG), and fasting insulin (FINS). And in the annotation section of the table, all abbreviations are fully explained to ensure their clarity.

7. We have corrected the incorrect words in Figure 7.

Reviewer #2: I have reviewed the authors' revised manuscript. In the process of revising the manuscript, the authors made revisions based on my suggestions and made responses to all the questions I raised. Therefore, I suggest that the journal accept this article.

Response to reviewer #2

Thanks for your hard work.

---

## [Decision Letter · Decision Letter 2]

2 Nov 2025

Dear Dr. Sui,

Thank you for submitting your manuscript to PLOS ONE. After careful consideration, we feel that it has merit but does not fully meet PLOS ONE’s publication criteria as it currently stands. Therefore, we invite you to submit a revised version of the manuscript that addresses the points raised during the review process.

We look forward to receiving your revised manuscript.

Kind regards,

José M. Alvarez-Suarez

Academic Editor

PLOS ONE

Journal Requirements:

Reviewers' comments:

Reviewer's Responses to Questions

**Comments to the Author**

Reviewer #1: (No Response)

2. Is the manuscript technically sound, and do the data support the conclusions?

Reviewer #1: Yes

3. Has the statistical analysis been performed appropriately and rigorously?

Reviewer #1: I Don't Know

4. Have the authors made all data underlying the findings in their manuscript fully available?

Reviewer #1: Yes

5. Is the manuscript presented in an intelligible fashion and written in standard English?

Reviewer #1: Yes

Reviewer #1: The revised manuscript has been improved, and just minor aspects should be checked.

- Material and methods: Section 2.7: Please, state clearly the number of cells seeded to perform each assay.

For the RT-qPCR, please, also include the micrograms of RNA transcribed and the nanograms of RNA amplified.

There are still certain grammatical and typo errors through the manuscript that require attention.

**Do you want your identity to be public for this peer review?** For information about this choice, including consent withdrawal, please see our Privacy Policy

Reviewer #1: No

---

## [Author Response · Author response to Decision Letter 3]

3 Nov 2025

Dear editor of the PLOS ONE,

We are submitting our revised manuscript entitled “Dietary flavonoids may improve insulin resistance: NHANES, network pharmacological analyses and in vitro experiments” to the PLOS ONE for your reconsideration of its suitability for publication. All authors have read and approved the manuscript. We have carefully taken the reviewers’ comments into account and provided responses to each of the points raised by the reviewers. Please refer to the present manuscript text.

Reviewer #1: The revised manuscript has been improved, and just minor aspects should be checked.

- Material and methods: Section 2.7: Please, state clearly the number of cells seeded to perform each assay.

For the RT-qPCR, please, also include the micrograms of RNA transcribed and the nanograms of RNA amplified.

There are still certain grammatical and typo errors through the manuscript that require attention.

Response to reviewer #1

1. Thank you very much for your suggestion. We have already listed in Section 2.7 the number of cells used in the experiment, as well as the micrograms of transcribed RNA and the nanograms of amplified RNA.

2. We have once again revised the inappropriate terms such as "hypertension" throughout the entire text.

---

## [Decision Letter · Decision Letter 3]

18 Nov 2025

Dietary flavonoids may improve insulin resistance: NHANES, network pharmacological analyses and in vitro experiments

PONE-D-25-21174R3

Dear Dr. Sui,

We’re pleased to inform you that your manuscript has been judged scientifically suitable for publication and will be formally accepted for publication once it meets all outstanding technical requirements.

Kind regards,

José M. Alvarez-Suarez

Academic Editor

PLOS ONE

Additional Editor Comments (optional):

Reviewers' comments:

Reviewer's Responses to Questions

**Comments to the Author**

Reviewer #1: All comments have been addressed

2. Is the manuscript technically sound, and do the data support the conclusions?

Reviewer #1: Yes

3. Has the statistical analysis been performed appropriately and rigorously?

Reviewer #1: I Don't Know

4. Have the authors made all data underlying the findings in their manuscript fully available?

Reviewer #1: Yes

5. Is the manuscript presented in an intelligible fashion and written in standard English?

Reviewer #1: Yes

Reviewer #1: All my queries have been answered. The manuscript could be accepted for its publication in the journal.

**Do you want your identity to be public for this peer review?** For information about this choice, including consent withdrawal, please see our Privacy Policy

Reviewer #1: No

---

## [Editor Report · Acceptance letter]

PONE-D-25-21174R3

PLOS ONE

Dear Dr. Sui,

I'm pleased to inform you that your manuscript has been deemed suitable for publication in PLOS ONE. Congratulations! Your manuscript is now being handed over to our production team.

Kind regards,

on behalf of

Professor José M. Alvarez-Suarez

Academic Editor

PLOS ONE